# RA-PbRL: Provably Efficient Risk-Aware Preference-Based Reinforcement Learning

**Yujie Zhao[1], Jose Efraim Aguilar Escamill[2], Weyl Lu[3], Huazheng Wang[2]**

[1] University of California, San Diego, [2] Oregon State University, [3] University of California, Davis

yuz285@ucsd.edu, aguijose@oregonstate.edu,
adslu@ucdavis.edu, huazheng.wang@oregonstate.edu

## Abstract

Reinforcement Learning from Human Feedback (RLHF) has recently surged in popularity, particularly for aligning large language models and other AI systems with human intentions. At its core, RLHF can be viewed as a specialized instance of Preference-based Reinforcement Learning (PbRL), where the preferences specifically originate from human judgments rather than arbitrary evaluators. Despite this connection, most existing approaches in both RLHF and PbRL primarily focus on optimizing a mean reward objective, neglecting scenarios that necessitate risk-awareness, such as AI safety, healthcare, and autonomous driving. These scenarios often operate under a one-episode-reward setting, which makes conventional risk-sensitive objectives inapplicable. To address this, we explore and prove the applicability of two risk-aware objectives to PbRL: nested and static quantile risk objectives. We also introduce Risk-Aware-PbRL (RA-PbRL), an algorithm designed to optimize both nested and static objectives. Additionally, we provide a theoretical analysis of the regret upper bounds, demonstrating that they are sublinear with respect to the number of episodes, and present empirical results to support our findings. Our code is available in https://github.com/aguilarjose11/PbRLNeurips.

## 1 INTRODUCTION

Reinforcement Learning (RL) [Russell and Norvig, 2010] is a fundamental framework for sequential decision-making, enabling intelligent agents to interact with and learn from unknown environments. This framework utilizes a reward signal to guide the selection of policies, where optimal policies maximize this signal. RL has demonstrated state-of-the-art performance in various domains, including clinical trials [Coronato et al., 2020], gaming [Silver et al., 2017], and autonomous driving [Basu et al., 2017].

Despite its performance, a significant limitation of the standard RL paradigm is the selection of a state-action reward function. In many real-world scenarios, constructing an explicit reward function is often a complex or unfeasible task. As a compelling alternative, Preference-based Reinforcement Learning (PbRL) [Busa-Fekete et al., 2014, Wirth et al., 2016] addresses this challenge by deviating from traditional quantifiable rewards for each step. Instead, PbRL employs binary preference feedback on trajectory pairs generated by two policies, which can be provided directly by human subjects. This method is increasingly recognized as a more intuitive and direct approach in fields involving human interaction and assessment, such as autonomous driving [Basu et al., 2017], healthcare [Coronato et al., 2020], and language models [Bai et al., 2022].A specialized and increasingly popular variant of PbRL is Reinforcement Learning from Human Feedback (RLHF)Bai et al. [2022], which has garnered considerable attention for aligning AI systems with human intentions—particularly in the domain of large-scale language models.

38th Conference on Neural Information Processing Systems (NeurIPS 2024).

Previous approaches to PbRL [Xu et al., 2020a, Coronato et al., 2020, Xu et al., 2020b, Chen et al., 2022, Zhan et al., 2023] mainly aim to maximize the mean reward or utility, which is risk-neutral. However, there is a growing need for risk-aware strategies in various fields where PbRL has shown empirical success. For example, in autonomous driving, PbRL reduces the computational burden by skipping the need to calculate reward signals for every state-action pair [Chen et al., 2022]. Despite this improvement, the nature of the problem makes dangerous actions costly. Thus, such risk-sensitive problem settings require risk awareness to ensure safety.

Risk-Aware PbRL also has implications for fields like generative AI [OpenAI, 2023, Chen et al., 2023], where harmful content generation remains a challenge for fine-tuning. In this scenario, a large language model (LLM) is often fine-tuned with user feedback by generating two prompt responses, A and B. Many approaches using RLHF [Ouyang et al., 2022] consider human feedback to penalize harmful content generation. Unfortunately, current approaches only minimize the average harmfulness of a response. This can be a challenge when responses are only harmful to a minority of users. Risk-aware PbRL tackles this challenge by directly aiming to decrease the harmfulness directly, rather than indirectly as with human feedback fine-tuning.

Despite substantial evidence highlighting the importance of risk awareness in PbRL, a significant gap persists in the theoretical analysis and formal substantiation of risk-aware PbRL approaches. This deficiency has spurred us to develop risk-aware measures and their theoretical analysis within PbRL. In standard RL, a variety of risk-aware measures have been explored, including a general family of risk-aware utility functions [Fei et al., 2020], iterated (nested) Conditional Value at Risk (CVaR) [Du et al., 2022], and risk-sensitive with quantile function form [Bastani et al., 2022]. In general, these measures can be categorized into two types: nested or static. Nested measures [Fei et al., 2020, Du et al., 2022] utilize MDPs to ensure risk sensitivity of the value iteration at each step under the current state, resulting in a more conservative approach. In contrast, static risk-aware measures Bastani et al., 2022 analyze the risk sensitivity of the whole trajectory's reward distribution. In developing and introducing risk-aware objectives in PbRL, we have encountered the following technical challenges in algorithm design and theoretical analysis:

**Rewards are defined over trajectories preference**   In PbRL, the reward function depends on the preference between two trajectories generated by the agent. We refer to this difference in how the reward function is computed as the one-episode-feedback characteristic. Consequently, the risk-aware objectives of standard RL like Du et al. [2022] and Fei et al. [2020] become unmeasurable since they depend on the state-action reward.

**Trajectory embedding reward assumption**   When computing the trajectory reward, it is assumed that an embedding mapping exists. By using the trajectory embedding along with some other vector embedding pointing towards high-rewarding trajectories, the reward is computed with a dot product. Unfortunately, the embedding mapping may not be linear. This means that the embedded trajectory vectors may not follow the Markovian assumption, making the embeddings history-dependent.

**Loss of linearity of Bellman function**   When using a quantile function to transform a risk-neutral PbRL algorithm into a risk-aware algorithm, the Bellman equation used to solve the problem becomes non-linear. This change to the bellman equation disrupts calculations on regret, making risk-neutral PbRL inapplicable. This is primarily due to the additional parameter $\alpha$, which modifies the underlying distribution.

In this paper, we address these challenges by studying the feasibility of risk-aware objectives in PbRL. We propose a provably efficient algorithm, Risk-Aware-PbRL(RA-PbRL), with theoretical and empirical results on its performance and risk-awareness. Our summary of contributions is as follows:

1. We analyze the feasibility of several risk-aware measures in PbRL settings and prove that in the one-episode-reward setting, nested and static quantile risk-aware objectives are applicable since they can be solved and computed uniquely in a given PbRL MDP.

2. We expand the state space in our formulation of a PbRL MDP and modify value iteration to address its history-dependent characteristics from the one-episode setting. These modifications enable us to use techniques like DPP to search for the optimal policy.

3. We develop a provably efficient (both computationally and statistically) algorithm, RA-PbRL, for nested and static quantile risk-aware objectives. To the best of our knowledge, we are the

first to formulate and analyze the finite time regret guarantee for a risk-aware algorithm with non-Markovian reward models for both nested and static risk-aware objectives. Moreover, we construct a hard-to-learn instance for RA-PbRL to establish a regret lower bound.

## 2 Related Work

### 2.1 Preference-based Feedback Reinforcement Learning

The incorporation of human preferences in RL, such as Jain et al. [2013], has been a subject of study for over a decade. This approach has proved to be successful and has been widely used in various applications, including language model training [Ouyang et al., 2022], clinical trials [Coronato et al., 2020], gaming [Silver et al., 2017], and autonomous driving [Basu et al., 2017]. PbRL can be categorized into three distinct types Wirth et al. [2017]: action preference, policy preference, and trajectory preference. Among these, trajectory preference is identified as the most general and widely studied form of preference-based feedback, as evidenced by the rich literature on the topic Chen et al. [2022], Xu et al. [2020a], Wu and Sun [2023]. As noted in our introduction, previous theoretical explorations on PbRL have predominantly aimed at achieving higher average rewards, which encompasses risk-neutral PbRL. We distinguish our work by taking the novel approach of formalizing the risk-aware PbRL problem.

### 2.2 Risk-aware Reinforcement Learning

In recent years, research on risk-aware RL has proposed various risk measures. Works such as Fei et al. [2020], Shen et al. [2014], Eriksson and Dimitrakakis [2019] integrate RL with a general family of risk-aware utility functions or the exponential utility criterion. Accordingly, studies like Bastani et al. [2022], Wu and Xu [2023] delve into the CVaR measure for the whole trajectory's reward distribution in standard RL. Further, Du et al. [2022] propose ICVaR-RL, a nested risk-aware RL formulation that addresses both regret minimization and best policy identification. Additionally, the work of Chen et al. [2023] presents an advancement in the form of a nested CVaR measure within the framework of RLHF. The limitation of this work lies in the selection of a random reference trajectory for comparison, causing an unavoidable linear strong nested CVaR regret. Consequently, we are left with only a preference equation from which we are unable to compute the state-action reward function for each step.

Practical and relevant trajectory or state-action embeddings are described in works such as [Pacchiano et al., 2021]. Therefore, the one-episode-reward might not even be sum-decomposable (the trajectory embedding details can be seen in sec.3.1 ). Compared to previous work, we use non-Markovian reward models that do not require estimating the reward at each step and explore both nested and static risk-aware objectives, aiming to provide a more general method.

## 3 Problem Set-up and Preliminary Analysis

### 3.1 PbRL MDP

We first define a modification of the classical Markov Decision Process (MDP) to account for risk: Risk Aware Preference-Based MDP (RA-PB-MDP). The standard MDP is described as a tuple, $\mathcal{M}(\mathcal{S}, \mathcal{A}, r_\xi^\star, \mathbf{P}^\star, H)$, where $\mathcal{S}$ and $\mathcal{A}$ represent finite state and action spaces, and $H$ denotes the length of episodes. Additionally, let $S := |\mathcal{S}|$ and $A := |\mathcal{A}|$ denote the cardinalities of the state and action spaces, respectively. $\mathbf{P}^\star : \mathcal{S} \times \mathcal{A} \to \mathcal{P}(\mathcal{S})$ is the transition kernel, where $\mathcal{P}(\mathcal{X})$ denotes the space of probability measures on space $\mathcal{X}$. A *trajectory* is a sequence

$$\xi_h \in \mathcal{Z}_h, \quad \mathcal{Z} = \bigcup_{h=1}^{H} \mathcal{Z}_h \quad \text{where} \quad \mathcal{Z}_h = (\mathcal{S} \times \mathcal{A})^{h-1} \times \mathcal{S}.$$

Intuitively, a trajectory encapsulates the interactions between an agent and the environment $\mathcal{M}$ up to step $h$. In contrast to the standard RL setting, where the reward function $r_h^\star(s_h, a_h)$ specifies the reward at each step $h$, a significant distinction in the PbRL MDP framework is that the reward function $r^\star$ is defined as $r_\xi^\star(\xi_H) : (\mathcal{S} \times \mathcal{A})^{H-1} \times \mathcal{S} \to [0, 1]$, denoting the reward of the entire trajectory.

**Reward model for the entire trajectory.** For any trajectory $\xi_H$, we assume the existence of a trajectory embedding mapping $\phi : \mathcal{Z}_H \to \mathbb{R}^{dim_\mathbb{T}}$, and the reward of the entire trajectory is defined as the function: $r_\xi^\star(\xi_H) := \langle \phi(\xi_H), \mathbf{w}_r^\star \rangle$. Here, $dim_\mathbb{T}$ denotes the trajectory embedding dimension. Finally, we denote $\phi(\xi_H) = (\phi_1(\xi_H), \dots, \phi_{dim_\mathbb{T}}(\xi_H))$.

**Assumption 3.1.** We assume that for all trajectories $\xi_H$ and for all $d \in \{1, \dots, dim_\mathbb{T}\}$, $\|\phi_d(\xi_H)\| \in \{0\} \cup [b, B]$ where $b, B > 0$ are known. $\|\mathbf{w}_r^\star\| \leq \rho_w$ and $\rho_w$ is known as well.

*Remark* 3.2. We assume the map $\phi$ is known to the learner. The sum of the state-action reward used in Chen et al. [2023] is one case of such map , where $\phi(\xi_H) = \sum_{h=1}^H \mathbb{I}(s_h, a_h)$ and $dim_\mathbb{T} = S \times A$. Therefore, for all $d \in \{1, \dots, dim_\mathbb{T}\}$, $\|\phi_d(\xi_H)\| \in \{0\} \cup \{1, \dots, H\}$

*Remark* 3.3. Assumption 3.1 implies that there is a gap between zero and some positive number $b$ in the absolute values of components of trajectory embeddings. This is evident for finite-step discrete action spaces, where we can enumerate all trajectory embeddings to find the smallest non-zero component, satisfying most application scenarios.

At each iteration $k \in [K]$, the agent selects two policies under a deterministic policy framework, $\pi_{1,k}$ and $\pi_{2,k}$, which generate two (randomized) trajectories $\xi_H^{1,k}$ and $\xi_H^{2,k}$. In PbRL, unlike standard RL where the agent receives rewards every step, the agent can only obtain the preference $o_k$ between two trajectories $\left(\xi_H^{1,k}, \xi_H^{2,k}\right)$. By making a query, we obtain a preference feedback $o_k \in \{0, 1\}$ that is sampled from a Bernoulli distribution:

$$o_k \sim Ber\left(\sigma\left(r_\xi^\star\left(\xi_H^{1,k}\right) - r_\xi^\star\left(\xi_H^{2,k}\right)\right)\right) \tag{1}$$

where $\sigma : \mathbb{R} \to [0, 1]$ is a monotonically increasing link function. We assume $\sigma$ is known like the popular Bradley-Terry model [Hunter, 2004], wherein $\sigma$ is represented by the logistic function. It is known that we can not estimate the trajectory reward without a known $\sigma$. Also, we assume $\sigma$ is Lipschitz continuous and $\kappa$ is its Lipschitz coefficient.

**History dependent policy.** Since the algorithm can only observe the reward for an entire episode until the end, it cannot make decisions based solely on the current state. The agent cannot observe the individual reward $r_h(s, a)$ and thus cannot compute the target value at each step. To circumvent this challenge, the algorithm should take action according to a history-dependent policy. A history-dependent policy $\Pi = \{\pi_h\}_{h \in [H]}$ is defined as a sequence of mappings $\psi_h : \mathcal{Z}_h \to \mathcal{A}$, providing the agent with guidance to select an action, given a trajectory $\xi_h \in \mathcal{Z}_h$ at time step $h$. For notation convenience, let $\Pi$ denote the set of all history-dependent deterministic policies.

### 3.2 Risk Measure

Because in PbRL we can only estimate the reward for the entire trajectory, the risk measure selected for PbRL must rely solely on the reward of the entire trajectory. That is, two trajectories with the same trajectory reward should contribute equally to the risk measure, even if their potential rewards at each step are different. Unlike Chen et al. [2023] that decomposes the reward at each step (where the solution is likely not unique) and then calculates the risk measure, this requirement ensures that the risk measure consistently and holistically reflects the underlying preference. We refer to risk measures that suitable for PbRL problems as *PbRL-risk-measures*. Here, we introduce two different risk-measures: nested and static quantile risk-aware measures, which are appropriate for PbRL-MDPs.

We first introduce the definition of quantile function and risk-aware objective. The quantile function of a random variable $X$ is $F_X^\dagger(\tau) = \inf\{x \in \mathbb{R} \mid F_X(x) \geq \tau\}$. We assume $F_X$ is strictly monotone, so it is invertible and we have $F_X^\dagger(\tau) = F_X^{-1}(\tau)$. The risk-aware objective is given by the Riemann-Stieljes integral:

$$\Phi(X) = \int_0^1 F_X^\dagger(\tau)\mathrm{d}G(\tau) \tag{2}$$

where $X$ is the random variable encoding the value of MDP, and $G$ is a weighting function over the quantiles. This class captures a broad range of useful objectives, including the popular CVaR objective [Bastani et al., 2022].

*Remark* 3.4. ($\alpha$-CVaR objective) Specifically, in $\alpha$-CVaR,

$$G(\tau) = \begin{cases} \frac{1}{\alpha}\tau & \text{if } \tau < \alpha, \\ 1 & \text{if } \tau \geq \alpha. \end{cases}$$

and $\Phi(X)$ becomes

$$\Phi(X) = \frac{1}{\alpha}\int_0^\alpha F_X^{-1}(\tau)\mathrm{d}\tau.$$

**Assumption 3.5.** *$G$ is $L_G$-Lipschitz continuous for some $L_G \in \mathbb{R}_{>0}$, and $G(0) = 0, G(1) = 1$.*

For example, for the $\alpha$-CVaR objective, we have $L_G = 1/\alpha$.

There are two prevalent approaches to Risk-aware-MDPs: *nested (or iterated)* (such as Iterated CVaR (ICVAR) [Du et al., 2022] and Risk-Sensitive Value Iteration (RSVI) [Fei et al., 2020]), and *static* (referenced in [Bastani et al., 2022, Wu and Xu, 2023]). MDPs characterized by an iterated risk-aware objective facilitate a value function and uphold a Bellman-type recursion.

**Nested PbRL-risk-measures.** For standard RL's MDP, the nested quantile risk-aware measure can be elucidated in Bellman equation type as follows:

$$\begin{cases} Q_h^\pi(s,a) & = r_h^\star(s,a) + \Phi\left(V_{h+1}^\pi(s'), s' \sim \mathbf{P}^\star(s,a)\right) \\ V_h^\pi(s) & = Q_h^\pi(s, \pi_h(s)) \\ V_{H+1}^\pi(s) & = 0, \quad \forall s \in \mathcal{S} \end{cases} \tag{3}$$

Here $r_h^\star(s,a)$ denotes the decomposed state-action reward in step $h$.

For the PbRL-MDP setting $\mathcal{M}(\mathcal{S}, \mathcal{A}, r_\xi^\star, \mathbf{P}^\star, H)$, the state-action's reward might not be calculated or the reward of the entire trajectory might not be decomposed. Therefore, the policy should be history-dependent. We rewrite the nested quantile objective's Bellman equation with the embedded trajectory reward as follows:

$$\begin{cases} \tilde{Q}_h^\pi(\xi_h, a) & = \Phi\left(\tilde{V}_{h+1}^\pi(s' \circ (\xi_h, a)), s' \sim \mathbf{P}^\star(s,a)\right), \\ \tilde{V}_h^\pi(\xi_h) & = \tilde{Q}_h^\pi(\xi_h, \pi_h(\xi_h)), \\ \tilde{V}_H^\pi(\xi_H) & = r^\star(\xi_H), \end{cases} \tag{4}$$

For any PbRL MDP $\mathcal{M}(\mathcal{S}, \mathcal{A}, r_\xi, \mathbf{P}, H)$, we use $\tilde{V}_h^{\pi, r_\xi, \mathbf{P}}(\xi_h)$ to denote the value iteration under the policy $\pi$, where $\pi$ is a history dependent policy.

**Lemma 3.6.** *For a given tabular MDP, the reward on the entire trajectory can be decomposed as $r_\xi^\star(\xi_H) = \sum_{h=1}^H r_h^\star(s_h, a_h)$, $V_1^\pi$ in Eq. 3 and $\tilde{V}_1^\pi$ in Eq. 4 are equivalent.*

The proof is detailed in Appendix B.1 due to space limitations.

**Static PbRL-risk-measures.** Standard MDPs with a static risk aware objective [Bellemare et al., 2017, Dabney et al., 2018] can be written in the distributional Bellman equation as follows:

$$Z_h^{(\pi)}(\xi_h) = \sum_{h'=h}^H r_{h'}^\star(s_{h'}, a_{h'}), \quad \xi_H \sim \mathbb{P}(\cdot \mid \Xi_h(\xi_H) = \xi_h)$$

$$F_{Z_h^{(\pi)}(\xi)}(x) = \sum_{s' \in \mathcal{S}} P(s' \mid S(\xi), \pi_h(\xi)) F_{Z_{h+1}^{(\pi)}(\xi \circ (s', \pi_h(\xi)))}(x - r_h^\star(s_h, a_h)) \tag{5}$$

$$V_1^\pi(s) = \int_0^1 F_{Z_1}^\dagger(\pi)(\tau) \cdot dG(\tau)$$

Where $S(\xi) = s$ for $\xi = (\dots, s)$ is the current state in trajectory $\xi$, $\Xi_h(\xi_H) = (s_1, a_1, s_2, a_2, \dots, s_{h-1}, a_{h-1}, s_h)$ denotes the first h steps' trajectory. $Z_h^{(\pi)}(\xi_h)$ denoted the reward from step $t$ given the current history. The *static reward* of $\pi$ is $Z_1^{(\pi)}(\xi)$, where $\xi = (s) \in \mathcal{Z}_1$ for $s \sim D$ is the initial history.

Also, we modify the distributional Bellman equation for PbRL MDP $\mathcal{M}(\mathcal{S}, \mathcal{A}, r_\xi^\star, \mathbf{P}^\star, H)$ settings as follows:

$$Z_h^{(\pi)}(\xi_h) = r_\xi^\star(\xi_H), \quad \xi_H \sim \mathbb{P}(\cdot \mid \Xi_h(\xi_H) = \xi_h)$$

$$F_{Z_h^{(\pi)}(\xi_h)}(x) = \sum_{s' \in \mathcal{S}} P(s' \mid S(\xi), \pi_h(\xi)) F_{Z_{h+1}^{(\pi)}(\xi \circ (s', \pi_h(\xi)))}(x) \tag{6}$$

$$\tilde{V}_1^\pi(s) = \int_0^1 F_{Z_1}^\dagger(\pi)(\tau) \cdot dG(\tau)$$

**Lemma 3.7.** *For a tabular MDP and a reward of the entire trajectory can be decomposed as* $r_\xi^\star(\xi_H) = \sum_{h=1}^H r_h^\star(s_h, a_h)$, $V_1^\pi$ *in Eq. 5 and* $\tilde{V}_1^\pi$ *in Eq. 6 are equivalent.*

The proof is detailed in Appendix B.2 due to space limitation.

Each of these risk measures possesses distinct advantages and limitations. Nested risk measures, which incorporate a Bellman-type recursion, can directly employ techniques such as the Dynamic Programming Principle (DPP) for computation. However, they are challenging to interpret and are not law-invariant [Hau et al., 2023]. On the other hand, static risk measures are straightforward to interpret, but the resulting optimal policy may not remain Markovian and becomes history-dependent. Consequently, techniques such as the DPP and the Bellman equation become inapplicable.

### 3.3 Objective

We define an optimal policy as:

$$\pi^\star \in \operatorname{argmax}_{\pi \in \Pi} \tilde{V}_1^\pi(s_1) \tag{7}$$

i.e., it maximizes the given objective for $\mathcal{M}$. $\tilde{V}_1^\pi(s_1)$ will be decided by the selected risk measure, where value iteration calculated using Eq. 4 and static calculated using Eq. 6.

**Assumption 3.8.** Regardless of nested or static CVaR objectives, we are given an algorithm for computing $\pi_\mathcal{M}^\star$ for a known $\mathrm{PbRL\text{-}MDP}$ $\mathcal{M}$.

A formal proof of Assumption 3.8 is given in Appendix F. When unambiguous, we drop $\mathcal{M}$ and simply write $\pi^\star$.

At the beginning of each episode $k \in [K]$, our algorithm $\mathfrak{A}$ chooses two policies $(\pi_{1,k}, \pi_{2,k}) = \mathfrak{A}(H_k)$, where $H_k = \{\xi_{1,k',H}, \xi_{2,k',H}, o_k\}_{k'=0}^k$ is the random set of episodes observed so far. Then, our goal is to design an algorithm $\mathfrak{A}$ that minimizes regret, which is naturally defined as:

$$\mathrm{Regret}(K) := \sum_{k=0}^K \left( 2\tilde{V}_1^{\pi^\star}(s_1) - \tilde{V}_1^{\pi_{1,k}}(s_1) - \tilde{V}_1^{\pi_{2,k}}(s_1) \right) \tag{8}$$

## 4 Risk Aware Preference based RL Algorithm

In this section, we introduce and analyze an algorithm called RA-PbRL for solving the PbRL problem with both nested and static risk aware objectives. Also, we establish a regret bound for it.

### 4.1 Algorithm

RA-PbRL is formally described in Algorithm 1. The development of RA-PbRL is primarily inspired by the PbOP algorithm, as delineated in Chen et al. [2022], which was originally proposed for risk-neutral PbRL environments. Building upon this foundation, one significant difference is how to choose a risk aware policy in estimated PbRL MDP, where the value iteration is different. We also use novel techniques to estimate the confidence set and explore for a policy, instead of using a bonus [Chen et al., 2022] (which is difficult to calculate in risk-aware problems) as in standard RL.

**Algorithm 1** RA-PbRL

**Require:** episode $K$, step $H$, initial state space $\mathcal{P}$, initial reward space $\mathcal{R}$, risk level $\alpha$, confidence parameter $\delta$

1: Set $\mathcal{B}_0^{\mathbf{P}} = \mathcal{P}, \mathcal{B}_0^r = \mathcal{R}$, Execute two arbitrary policies $\pi_{1,0}$ and $\pi_{2,0}$ for one episode, respectively, and then observe the trajectory $\tau_{1,0}$ and $\tau_{2,0}$ and the preference $o_0$.

2: **for** $k = 1 \cdots K$ **do**

3:     Calculate the probability estimation $\hat{\mathbf{P}}_k$:

$$\hat{\mathbf{P}}_k = \arg\min_{\mathbf{P} \in \mathcal{P}} \sum_{i=1}^{2} \sum_{k'=0}^{k-1} \sum_{h=1}^{H} |\langle \mathbf{P}(s_{i,k',h}, a_{i,k',h}), \mathbb{I}(s_{i,k',h+1}) \rangle|^2.$$

4:     Update transition confidence set :

$$\mathcal{B}_k^{\mathbf{P}} = \left\{ \mathbf{P}' \mid \sum_{s' \in \mathcal{S}} \left| \hat{\mathbf{P}}^k(s' \mid s, a) - \mathbf{P}'(s' \mid s, a) \right| \leq \sqrt{\frac{2S \log\left(\frac{2KHSA}{\delta}\right)}{n_k(s,a)}} \right\} \bigcap \mathcal{B}_{k-1}^{\mathbf{P}}$$

5:     Calculate the reward estimation:

$$\hat{r}_k(\cdot) = \arg\min_{r \in \mathcal{R}} \sum_{k'=0}^{k-1} \left( \sigma(r(\tau_{1,k'}) - r(\tau_{2,k'})) - o_{k'} \right)^2$$

6:     Update the confidence set:

$$\mathcal{B}_k^r = \left\{ r'(\cdot) \left| \sum_{k'=0}^{k-1} \left[ \sigma(\hat{r}_k(\tau_{1,k'}) - \hat{r}_k(\tau_{2,k'})) - \sigma(r'(\tau_{1,k'}) - r'(\tau_{2,k'})) \right]^2 \leq \beta_{r,k}(\delta) \right. \right\} \bigcap \mathcal{B}_{k-1}^r$$

7:     Update policy confidence set:

$$\Pi_k = \{ \pi \mid \max_{r_\xi \in \mathcal{B}_k^r, \mathbf{P} \in \mathcal{B}_k^{\mathbf{P}}} (\tilde{V}_{1,r_\xi,\mathbf{P}}^{\pi}(s_1) - \tilde{V}_{1,r_\xi,\mathbf{P}}^{\pi'}(s_1)) \geq 0, \forall \pi' \} \bigcap \Pi_{k-1}$$

8:     Compute $(\pi_{1,k}, \pi_{2,k})$:

$$(\pi_{1,k}, \pi_{2,k}) = \arg\max_{\pi_1, \pi_2 \in \Pi_k} \max_{r \in \mathcal{B}_k^r, \mathbf{P} \in \mathcal{B}_k^{\mathbf{P}}} (\tilde{V}_{1,r_\xi,\mathbf{P}}^{\pi_1}(s_1) - \tilde{V}_{1,r_\xi,\mathbf{P}}^{\pi_2}(s_1))$$

9:     Observe the trajectory $\xi_{1,k,H}, \xi_{2,k,H}$, and the preference $o_k$

10:    Calculate the state-action visiting time before episode $k$: $n_k(s,a)$

11: **end for**

**The overview of the algorithm**. Now we introduce the main part of our algorithm. In line 1, we initialize the transition kernel function and reward function confidence set, and execute two arbitrary policies at first. For every episode, we observe history samples and accordingly estimate the transition kernel function (line 3) and update its confidence set (line 4) as well as the reward function (line 5) and its confidence set (line 6). Both estimation and calculation used the standard least-squares regression. Based on the confidence sets, we maintain a policy set in which all policies are near-optimal with minor sub-optimality gap with high probability in line 7. In line 8, we execute the most exploratory policy pair in the policy set and observe the preference between the trajectories sampled using these two policies.

**The key difference between nested and static objective.** The estimation of the transition kernel (line 4 in Algorithm 1) and the construction of confidence set (line 6 in Algorithm 1) are similar for both nested and static objectives. The difference lies in the value iteration, which is defined in Eq. 4 for nested objective and Eq. 6 for static objective. The bounds for regrets are different since the estimation error's impact is different as we are going to show below.

### 4.2 Analysis

**Theorem 4.1 ( Nested object regret upper bound).** *With at least probability $1 - \delta$, the nested quantile risk aware object regret of RA-PBRL is bounded by:*

$$\mathrm{Reg}_{nested}(K)$$

$$\leq \mathcal{O}\left( L_G H^{\frac{3}{2}} \sqrt{K} SA \log\left(\frac{KHSA}{\delta}\right) \cdot \frac{1}{\sqrt{\min_{\pi,h,s:\omega_{\pi,h}(s)>0} \omega_{\pi,h}(s)}} \right)$$

$$+ \mathcal{O}\left( \frac{B}{\kappa b} dim_{\mathbb{T}} \sqrt{\log\left(\frac{K dim_{\mathbb{T}}}{\delta}\right) \log\left(\frac{K(1+2B\rho_w)}{\delta}\right)} \frac{1}{\min_{\pi,d} \omega_{dim,\pi}(d)} \right) \quad (9)$$

*Where $w_{\pi,h}(s)$ denotes the probability of visiting state-action pair at h th step with policy $\pi$ and $\min_{\pi,d} \omega_{dim,\pi}(d)$ denotes probability of trajectory $\xi_H$'s d th feature $\Phi_d(\xi_H) \neq 0$ with the policy $\pi$.*

The proof of this theorem is provided in Appendix D.20. The first term of the regret arises from the estimation error of the transition kernel, primarily dominated by $\min_{\pi,h,s:w_{\pi,h}(s)>0} w_{\pi,h}(s)$. The second term is due to the estimation error of the trajectory reward weights, significantly impacted by $min_{\pi,d}\omega_\pi(d)$ . In fact, these factors are unavoidable in the lower bound in certain challenging cases. Thus, they characterize the inherent problem difficulty, i.e., in achieving the nested risk-aware objective, the agent will be highly sensitive to some state-actions or features that are difficult to observe and require substantial effort to explore. This may result in inefficiency in many scenarios.

.

**Theorem 4.2** (**Static object regret upper bound**). *The static quantile risk aware object regret of RA-PBRL is bounded by:*

$$\text{Reg}_{static}(K)$$

$$\leq \mathcal{O}\left(L_G S^2 A H^{\frac{3}{2}} \sqrt{K} log\left(K/\delta\right)\right) + \mathcal{O}\left(L_G dim_{\mathbb{T}} \sqrt{K \log(KB\rho_w) \log\left(\frac{K\left(1+2B\rho_w\right)}{\delta}\right)}\right)$$

$$\tag{10}$$

The proof of this theorem is provided in Appendix D.21. Notice that the regret for both the nested risk objective and the static risk objective of Algorithm 1 are sublinear with respect to $K$, making RA-PbRL the first provably efficient algorithm with one-episode-reward for these two objectives. Additionally, compared to Chen et al. [2023], we achieve the goal of having both policies gradually approach optimality. Moreover, in comparison to the nested risk-aware objective, the static objective focuses on the risk measure of the entire distribution, primarily influenced by the Lipschitz coefficient $L_G$ of the quantile function and is less constrained by certain specific cases.

**Theorem 4.3** (**Nested object regret lower bound**). *The nested quantile risk aware object regret of RA-PBRL is bounded by:*

$$\text{Regret}(K) \geq \mathcal{O}\left(\min\left\{B\rho_w \sqrt{\frac{AK}{\min_{\pi,h,s:p_{\pi,h}(s)>0} w_{\pi,h}(s,a)}}, B\sqrt{\frac{AK}{\min_{\pi,d} \omega_{dim,\pi}(d)}}\right\}\right) \tag{11}$$

We provide our proof in E.1 by two hard-to-learn constructions. By the two instances, we show that the two factors, $\min_{\pi,h,s:p_{\pi,h}(s)>0} w_{\pi,h}(s)$, $\min_{\pi,d} \omega_{dim,\pi}(d)$ , are unavoidable in some cases.

**Theorem 4.4** (**Static object regret lower bound**). *The static quantile risk aware object regret of RA-PBRL is bounded by:*

$$\text{Regret}(K) \geq \mathcal{O}(S^2 A + dim_{\mathbb{T}})\sqrt{K}$$

The proof of this theorem is similar to Theorem 4.5 in Chen et al. [2022].

## 5 Experiment Results

In this section, we assess the empirical performance of RA-PbRL (Algorithm 1). For a comparative analysis, we select two baseline algorithms: PbOP, as described in Chen et al. [2022], which is a PbRL algorithm utilizing general function approximation, and ICVaR-RLHF, detailed in Chen et al. [2023], which is a risk-sensitive Human Feedback RL algorithm. These baselines represent the most closely aligned algorithms with RA-PbRL, especially in terms of employing general function approximation. The evaluation of empirical performance is conducted through the lens of static regret, as defined in Eq. 8.

### 5.1 Experiment settings: MDP

In our experimental framework, we configure a straightforward tabular MDP characterized by finite steps $H = 6$, finite actions $A = 3$, state space $S = 4$, and risk levels $\alpha \in \{0.05, 0.10, 0.20, 0.40\}$. For each configuration and algorithms, we perform 50 independent trials and report the mean regret across these trials, along with 95% confidence intervals. The outcomes are depicted in Figures 1 and 3, where the solid lines represent the empirical means obtained from the experiments, and the width of the shaded regions indicates the standard deviation of the experiments.

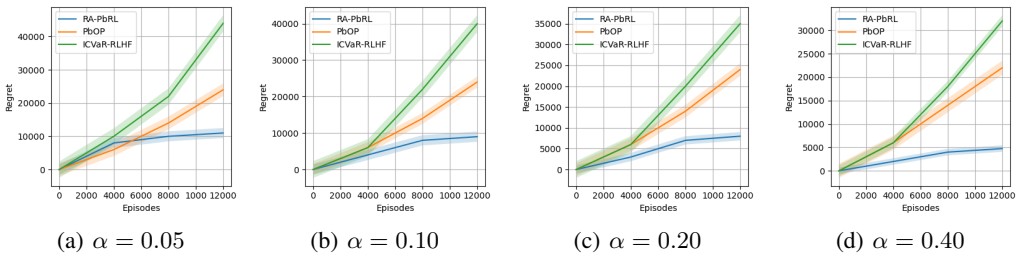

(a) $\alpha = 0.05$  (b) $\alpha = 0.10$  (c) $\alpha = 0.20$  (d) $\alpha = 0.40$

Figure 1: Cumulative regret for static CVaR over different $\alpha$

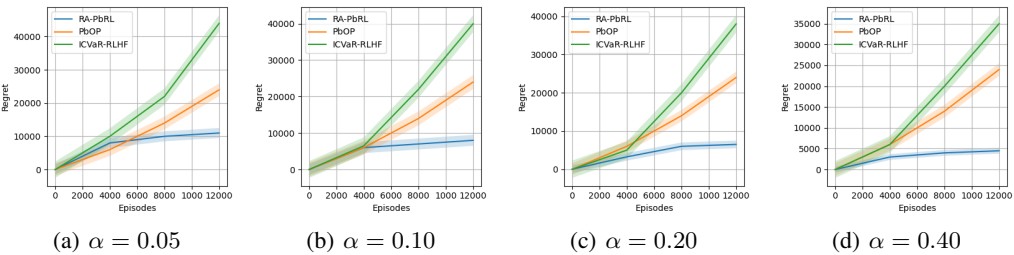

(a) $\alpha = 0.05$  (b) $\alpha = 0.10$  (c) $\alpha = 0.20$  (d) $\alpha = 0.40$

Figure 2: Cumulative regret for nested CVaR over different $\alpha$.

## 5.2 Experiment settings: Half Cheetah

Aditionally, we implement our algorithm to solve MuJoCo's Half-cheetah simulation. We implement our proposed algorithm alongside PbOP and TRC Kim and Oh [2022]. The objective of the algorithms is to learn to control a robot simulation of a cheetah to run forward. This problem differs from the previous setting in that it is more challenging and uses a continuous state-action space. Because the algorithm ws originally implemented for discrete state-action spaces, we assume the transition, reward, and policy functions can be parameterized by a linear function, which is optimized for using gradient descent.

Similarly to the previous experiment, we run both policies in the MuJoCo setting until 1,000 timesteps have passed. We repeat this for 100 episodes, saving the interactions and final preferences based on the cumulative reward after each episode. Finally, we use the data to perform gradient descent for a pre-defined number of repetitions. We repeat the cycle of interaction and training another 100 times. In total, the algorithm sees 10,000,000 timesteps and 10,000 preference reward signals.

The key idea behind the implementation of our algorithm lies in the initial optimization of the learned transition and reward functions using the data collected during the interaction. We iterate through tuples containing the initial state, action taken, and transitioned state. We perform stochastic gradient descent to find the best vectors that parameterize the transition and reward functions to predict the collected data.

After obtaining the best transition and reward functions, we use their parameterization to create a new parameterization of the value function in line 8. In our case, we simply concatenate the vectors parameterizing the transition and reward functions alongside a vector parameterizing the policies. The policy vector used depends on the policy being optimized (the best or exploratory policies.) We then compute the $\alpha$-CVaR over the preferences obtained using the final cumulative reward. We then optimize the value function parameterization using this as the training data, and perform stochastic gradient descent. To follow the theoretical bounds we establish, we compute the distance between the initial transition and reward parameterizations used in the value function, rolling back the parameterization if the distance is larger than what is established by the theoretical bounds.

## 5.3 Experimental results

As depicted in Figure 1 and 3, the regret of RA-PbRL over static and nested CVaR initially exhibits a linear growth with respect to $K$, transitioning to sublinear growth upon reaching a certain threshold. This behavior aligns with the conclusions drawn in Section 4.2. It is important to note that increased

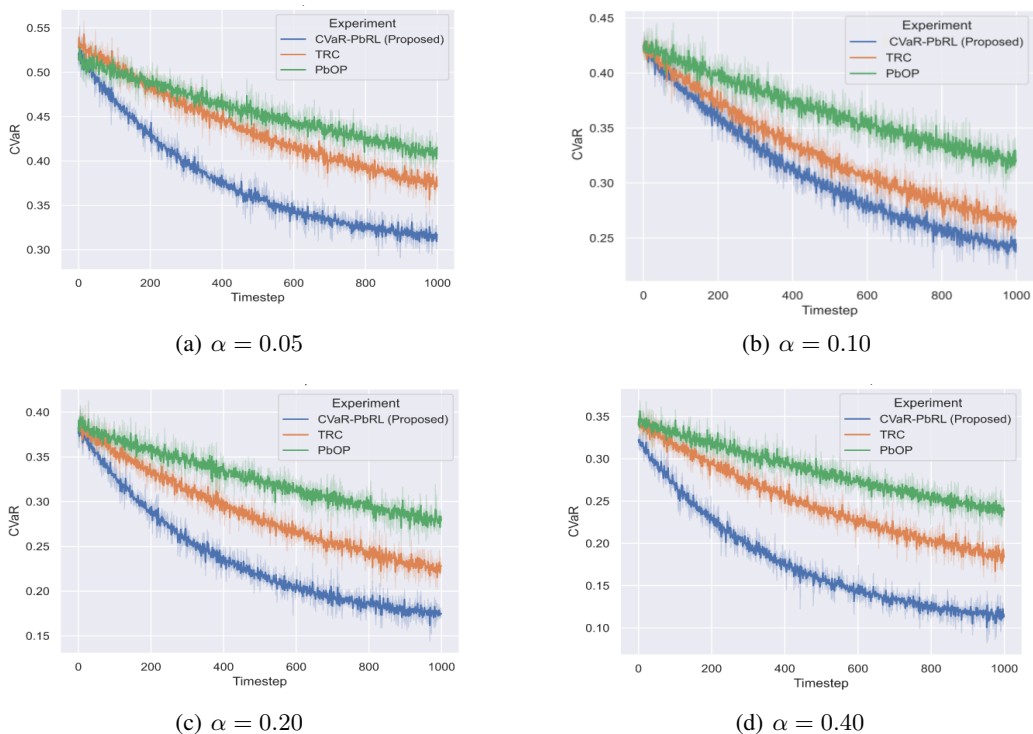

(a) $\alpha = 0.05$                  (b) $\alpha = 0.10$

(c) $\alpha = 0.20$                  (d) $\alpha = 0.40$

Figure 3: Cumulative regret for static CVaR in the MuJoCo setting over different $\alpha$.

risk aversion ($\alpha \to 0$) introduces greater uncertainty, as evidenced by the larger variance regions observed in the experiments. Notably, the regret of the bad-scenarios associated with RA-PbRL is significantly lower compared to those of other algorithms. It can additionally be observed that as $\alpha$ is increased, the regret improves, which is expected as riskier behavior can also improve the odds of finding useful behaviors.

## 6 Conclusion

In this paper, we investigate a novel PbRL algorithm for solving problems requiring risk awareness. We explore static and nested measures to introduce risk awareness to PbRL settings. To the best of our knowledge, our proposed RA-PbRL algorithm is the first provably efficient Preference-based Reinforcement Learning (PbRL) algorithm that incorporates both nested and static risk objectives in one algorithm. Our algorithm is built on innovative techniques for the efficient approximation of regret. A core finding in our investigation is the strong influence of the state and trajectory dimensions with respect to the nested risk objective regret. On the other hand, the static risk objective regret is mainly determined by the quantile function.

We have also identified the following four limitations to our work. (1) Our comparison feedback is limited to two trajectories. An interesting, more general approach could consider n-wise comparisons. (2) The reward functions are assumed to be linear in this work for the sake of simplicity. (3) Although this work has considered more general risk measures, we have still made certain assumptions that limit the generality of our results. (4) There is still room for future improvements to further close the gap between upper and lower bounds. We believe this work opens several avenues for future research, including establishing the concrete lower bounds of risk-aware PbRL, improving the computational complexity of the algorithm, and conducting experiments in more diverse and interesting environments/simulations.

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

# A   Notation

We clarify the notations that appear uniquely in this paper to avoid confusion.

| variable with $^\star$ | Ground truth of the variable. |
|---|---|
| $\mathcal{S}$, $\mathcal{A}$ | Finite state space, action space. |
| $s$, $a$ | State, action. |
| $S$, $A$ | Dimension of state space, action space. |
| $\mathbf{P}$ | Probability transition kernel. |
| $H$ | Length of episode. |
| $\mathcal{Z}_h$ | Set of all h-step trajectories. |
| $\xi_h$ | A trajectory contains $h$ steps. |
| $S_h(\xi)$ | State of $h$-th step of a trajectory. |
| $\Xi_h(\xi_{h'})$ | $1 \sim h$ steps trajectory of a $h'$-step trajectory. |
| $r_h(s_h, a_h)$ | Reward function of a single step $h$. |
| $r_\xi(\xi_H)$ | Reward function of a whole trajectory. |
| $r_{1:h}$ | Reward of the $1 \sim h$ steps of a trajectory. |
| $V_h^\pi(s)$ | Value function of state $s$ at step $h$ under policy $\pi$ in normal MDP. |
| $\tilde{V}_h^\pi(s)$ | Value function of state $s$ at step $h$ under policy $\pi$ in PbRL-MDP. |
| $\Phi(X)$ | Riemann-Stieljes integral of over a random variable $X$, i.e. the risk-aware objective. |
| $N\left(\mathcal{F}_\mathcal{R}, \frac{1}{K}, \|\cdot\|_\infty\right)$ | Covering number of function class $\mathcal{F}_\mathcal{R}$ at scale $1/K$ under $\|\cdot\|_\infty$ norm. |

# B   Risk Aware Object Computability

Unlike standard RL where each step's reward can be observed, PbRL represents a type of RL characterized by once-per-episode feedback. As a result, our observable and estimable parameters are confined to the trajectory reward $r_\xi^\star$ and transition probability functions $\mathbf{P}^\star$. Consequently, the traditional risk-aware objective might be unsuitable, as the reduction in available information prevents the computation of the original risk-aware measure. The risk measure selected for PbRL must satisfy the following condition: it should remain unique across MDPs where policies, trajectory rewards and transition probability functions even when each trajectory is fixed, but different step rewards, $r^\star(s,a)$, vary. This requirement ensures that the risk measure consistently reflects the underlying preferences regardless of variations in specific step rewards.

## B.1   Nested Object

**Theorem B.1.** *For the tabular MDP and the reward of the entire trajectory can be decomposed as $r_\xi^\star(\xi_H) = \sum_{h=1}^H r_h^\star(s_h, a_h)$, $V_1^\pi$ in Eq. 5 and $\tilde{V}_1^\pi$ in Eq.6 are equivalent.*

*Proof.* Firtly, according to Givan et al. [2003], Lowd and Davis [2010], any tabular MDP can be reformulated as a decision tree-like MDP. Thus, considering a tree-like structure for an MDP implies the following characteristics:

1. The state transition graph of the MDP is connected and acyclic. 2. Each state in the MDP corresponds to a unique node in the tree. 3. There is a single root node from which every other node is reachable via a unique path. 4. The transition probabilities between states follow the Markov property, i.e., the probability of transitioning to any future state depends only on the current state and not on the sequence of events that preceded it.

Formally, let $S$ be the set of states and $p_{ij}$ the transition probabilities between states $s_i$ and $s_j$. The transition matrix $P$ for an MDP with a tree-like structure is defined such that:

$$p_{ij} > 0 \text{ if there is an edge between } s_i \text{ and } s_j \text{ in the tree, and } p_{ij} = 0 \text{ otherwise.}$$

Moreover, for each non-root node $s_j$, there exists exactly one $s_i$ such that $p_{ij} > 0$, and $s_i$ is the unique parent of $s_j$ in the tree structure.

To classify the two value iteration in Eq. 3 and Eq. 4, we denote the value given by Eq. 4 as $\tilde{V}_h^\pi$ and the value given by Eq. 3 as $V_h^\pi$, thus, in tabular tree-like MDP with the reward of the entire trajectory

which can be decomposed as $r_\xi^\star(\xi_H) = \sum_{h=1}^H r_h^\star(s_h, a_h)$, we have the following relationship:

$$\tilde{V}_h^\pi = V_h^\pi + r_{1:h-1}^\star$$

where $r_{1:h}$ denotes Reward of the $1 \sim h$ steps of a trajectory. We prove this relationship by mathematical induction.

**Initial case.** Using the tree-like PbRL-MDP algorithm and the initial conditions of the Bellman equation, at the final step $h = H$, we have

$$\tilde{V}_H^\pi = r_H^\star(s_H', \pi(\xi_{H-1})) + r_{1:H-1}^\star \tag{12}$$
$$= V_H^\pi + r_{1:H-1}^\star \tag{13}$$

**Induction step.** We now proved that if

$$\tilde{V}_{h+1}^\pi = V_{h+1}^\pi + r_{1:h}^\star$$

holds, then

$$\tilde{V}_h^\pi = V_h^\pi + r_{1:h-1}^\star$$

also holds.

Since this tree-like MDP's policy $\pi$ is fixed, it has only one path to arrive h th state ($s_h$), denoted as:

$$\Xi_h(\xi_{H,1}) = \Xi_h(\xi_{H,2}) \quad \forall \xi_{H,1}, \xi_{H,2} \in \{\xi_H \mid S_h(\xi_H) = s_h\} \tag{14}$$

Therefore, $r_{1:h-1}^\star$ is unique.

$$\tilde{V}_h^\pi = \Phi\left(V_{h+1}^\pi(s_{h+1}') + r_{1:h}^\star\right), \quad s_{h+1}' \sim \mathbf{P}^\star(s, a) \tag{15}$$
$$= \Phi\left(V_{h+1}^\pi(s_{h+1}') + r_h^\star(s_h, \pi_h(\xi_h)) + r_{1:h-1}^\star\right), \quad s_{h+1}' \sim \mathbf{P}^\star(s, a) \tag{16}$$
$$= \Phi\left(V_{h+1}^\pi(s_{h+1}') + r_h^\star(s_h, \pi_h(\xi_h))\right) + r_{1:h-1}^\star, \quad s_{h+1}' \sim \mathbf{P}^\star(s, a) \tag{17}$$
$$= V_h^\pi + r_{1:h-1}^\star \tag{18}$$

By applying conclusion, we observe that when $h = 1$

$$\tilde{V}_1^\pi = V_1^\pi.$$

Thus, we have proven that the for the tabular MDP and the reward of the entire trajectory can be decomposed as $r_\xi^\star(\xi_H) = \sum_{h=1}^H r_h^\star(s_h, a_h)$, $V_1^\pi$ in Eq. 3 and Eq. 4 are equivalent.

$\square$

## B.2 Static Object

**Lemma B.2.** *For the tabular MDP and the reward of the entire trajectory can be decomposed as* $r_\xi^\star(\xi_H) = \sum_{h=1}^H r_h^\star(s_h, a_h)$, $V_1^\pi$ *in Eq. 5 and* $\tilde{V}_1^\pi$ *in Eq.6 are equivalent.*

*Proof.* To classify the two value iteration in Eq. 5 and Eq. 6, we denote the value given by Eq. 5 as $\tilde{V}_h^\pi$ and the value given by Eq. 6 as $V_h^\pi$, thus, in tabular tree-like MDP with the reward of the entire trajectory which can be decomposed as $r_\xi^\star(\xi_H) = \sum_{h=1}^H r_h^\star(s_h, a_h)$, we have the following relationship:

$$\tilde{V}_h^\pi = V_h^\pi + r_{1:h-1}^\star$$

where $r_{1:h}$ denotes Reward of the $1 \sim h$ steps of a trajectory.

Now, We prove this relationship.

Since this tree-like MDP's policy $\pi$ is fixed, it has only one path to arrive h th state ($s_h$), denoted as:

$$\Xi_h(\xi_{H,1}) = \Xi_h(\xi_{H,2}) \quad \forall \xi_{H,1}, \xi_{H,2} \in \{\xi_H \mid S_h(\xi_H) = s_h\} \tag{19}$$

Therefore, $r^\star_{1:h-1}$ is unique. By definition,

$$\tilde{V}^\pi_h = r^\star_\xi(\xi_H) \quad \xi_H \sim \mathbb{P}\left(\cdot \mid \Xi_h(\xi_H) = \xi_h\right) \tag{20}$$

$$= r^\star_\xi(\Xi_h(\xi_H)) + r^\star_{1:h-1}, \quad \xi_H \sim \mathbb{P}\left(\cdot \mid \Xi_h(\xi_H) = \xi_h\right) \tag{21}$$

$$= V^\pi_h + r^\star_{1:h-1} \tag{22}$$

By applying conclusion, we observe that when $h = 1$

$$\tilde{V}^\pi_1 = V^\pi_1.$$

Thus, we have proven that the for the tabular MDP and the reward of the entire trajectory can be decomposed as $r^\star_\xi(\xi_H) = \sum_{h=1}^{H} r^\star_h(s_h, a_h)$, $V^\pi_1$ in Eq. 5 and $\tilde{V}^\pi_1$ in Eq.6 are equivalent.

$\square$

## C  Difference between nested and static risk measure

To explain the difference between nested and static risk measure, we present a simple example that demonstrates their characters.

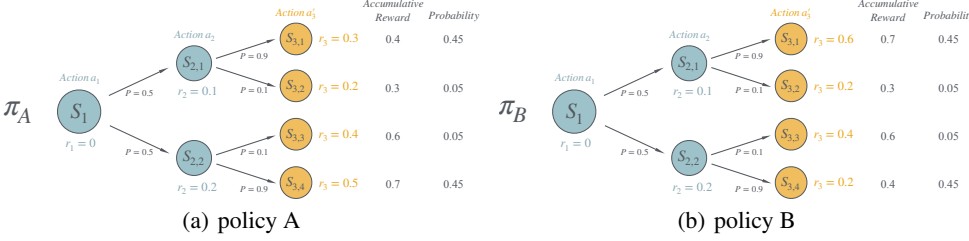

Figure 4: Cumulative regret for the different $\alpha$

First, we construct a MDP instance in fig. 4 where two policies exhibit identical reward distributions and consequently demonstrate equivalent preference will be observed . However, within this instance, these policies yield different outcomes under the nested and static CVaR metrics.

The state space is $\mathcal{S} = \{S_1, S_{2,1}, S_{2,2}, S_{3,1}, S_{3,2}, S_{3,3}, S_{3,4}\}$, where $S_1$ is the initial state.

The policy space is $\Pi = \{\pi_A, \pi_B\}$. Both policy have have the same action in the first step $(a_1)$ and second step $(a_2)$, but have the different action in the third step $(a_3, a'_3)$.

The reward functions are as follows. $r(S_1, a_1) = 0$, $r(S_{2,1}, a_2) = 0.1$, $r(S_{2,2}, a_2) = 0.2$, $r(S_{3,1}, a_3) = 0.3$, $r(S_{3,2}, a_3) = 0.2$, $r(S_{3,3}, a_3) = 0.4$, $r(S_{3,2}, a_3) = 0.5$, $r(S_{3,1}, a'_3) = 0.6$, $r(S_{3,2}, a'_3) = 0.2$, $r(S_{3,3}, a'_3) = 0.4$, $r(S_{3,2}, a'_3) = 0.2$.

The transition distributions are as follows. $P(S_{2,1} \mid S_1, a_1) = 0.5$, $P(S_{2,2} \mid S_1, a_1) = 0.5$, $P(S_{3,1} \mid S_{2,1}, a_2) = 0.1$, $P(S_{3,2} \mid S_{2,1}, a_2) = 0.9$, $P(S_{3,3} \mid S_{2,2}, a_2) = 0.1$, $P(S_{3,4} \mid S_{2,2}, a_2) = 0.9$.

As depicted on the right side of the figure, the distribution of rewards is consistent. Consequently, the human feedback preferences for the two policies are identical. We list the differing risk measures for these two policies in Table C.

| Metric | $\alpha$ | Value($\pi_A$) | Value($\pi_B$) |
|---|---|---|---|
| Nested CVaR | 0.2 | 0.33 | 0.36 |
| Static CVaR | 0.2 | 0.41 | 0.41 |

## D  REGRET UPPER BOUND FOR ALGORITHM 1

In this section, we present the full proof of Assumption 1's regret upper bound. The proof consists of parts.

## D.1 REWARD ESTIMATION ERROR

**Lemma D.1.** *Reward confidence set construction. Fix $\delta \in (0, 1)$, with probability at least $1 - \delta$, for all $k \in [K]$,*

$$\sum_{k'=1}^{k-1} [\sigma(\widehat{r}_k(\xi_1) - \widehat{r}_k(\xi_2)) - \sigma(r^\star(\xi_1) - r^\star(\xi_2))]^2 \leq \beta_{r,k} \tag{23}$$

*where*

$$\beta_{r,k}\left(\delta, \frac{1}{K}\right) \leq \mathcal{O}\left(dim_{\mathbb{T}} log\left(K\left(1 + 2B\rho_w\right)\right) + log(\frac{1}{\delta})\right) \tag{24}$$

*Proof.* This lemma can be proved by the direct application of Lemma G.5. Let $X_t = (\xi_{t,1}, \xi_{t,2})$ and $Y_t = o_t$, and $\mathcal{F}_{r,t} = \{f_r | f_r(\cdot, \cdot) = \sigma(r(\cdot) - r(\cdot))\}$. Then, we have that $X_t$ is $\mathcal{F}_{t-1}$ measurable and $Y_t$ is $\mathcal{F}_t$ measurable. According to Hoeffding's inequality (Theorem G.8), $\{Y_t - f_r(X_t)\}$ is $\frac{1}{2}$-sub-gaussian conditioning, and $\mathbb{E}[Y_t - f_r(X_t) \mid \mathcal{F}_{t-1}] = 0$. By Lemma G.5 and G.4, since the linear trajectory embedding function $\phi : \mathcal{Z}_H \to \mathbb{R}^{dim_{\mathbb{T}}}$, with probability at least $1 - \delta$, we have

$$\beta_{r,k}\left(\delta, \frac{1}{K}\right) \leq \mathcal{O}\left(dim_{\mathbb{T}} log\left(K\left(1 + 2B\rho_w\right)\right) + log(\frac{1}{\delta})\right) \tag{25}$$

$\square$

**Lemma D.2.** *Reward estimation error of trajectory embedding. For any $k \in [0, \ldots, K]$, reward confidence set $\mathcal{B}_k^r$, where the reward function embedding weight can be noted as $\mathbf{w}_r = (w_1, \ldots, w_{dim_{\mathbb{T}}})$, any fixed trajectory $\xi_H \in \mathcal{Z}_H$, trajectory embedding $\phi(\xi_H) = (\phi_1, \ldots, \phi_{dim_{\mathbb{T}}})$, with probability at least $1 - \delta$, it holds that,*

$$max_{r_1, r_2 \in \mathcal{B}_k^r} |(w_{r_1,d} - w_{r_2,d})| \leq \mathcal{O}\left(\frac{1}{\kappa b}\sqrt{\frac{dim_{\mathbb{T}} log\left(K\left(1 + 2B\rho_w\right)\right) + log(\frac{1}{\delta})}{n_{dim,K}(d)}}\right) \tag{26}$$

*where $n_{\xi,k}$ denotes the number of times $\xi_H$ was visted up to episode $k$.*

*Proof.* According to Lemma D.1 and the assumption of link function, for fixed $k \in [0, \ldots, K]$, and $d \in [0, \ldots, dim_{\mathbb{T}}]$, we have,

$$max_{r_1, r_2 \in \mathcal{B}_K^r} \sum_{k'=1}^{k} |(w_{r_1,d} - w_{r_2,d})|^2 \, b^2 \mathbb{I}(B \neq 0) \tag{27}$$

$$\leq \sum_{k=1}^{K} |((w_{r_1,d} - w_{r^\star,d}) \cdot B)|^2 \tag{28}$$

$$\leq \frac{\beta_{r,K}}{\kappa^2} \tag{29}$$

Where $n_{dim,k}(d)$ denotes the number of $\phi_d(\xi_H) \neq 0$ among $1 \sim k$ episode's trajectory.

Using Cauchy-Schwarz inequality and Lemma G.6, we have,

$$\max_{r_1, r_2 \in \mathcal{B}_k^r} |(w_{r_1,d} - w_{r_2,d})| \leq \mathcal{O}\left(\frac{1}{\kappa b}\sqrt{\frac{dim_{\mathbb{T}} log\left(K\left(1 + 2B\rho_w\right)\right) + log(\frac{1}{\delta})}{n_{dim,K}(d)}}\right) \tag{30}$$

$\square$

**Lemma D.3.** *Reward estimation error of the whole distribution.*

*For $b_k^r(\xi_1, \xi_2) = \max_{r_1, r_2 \in \mathcal{B}_k^r}[(r_1(\xi_1) - r_1(\xi_2)) - (r_2(\xi_1) - r_2(\xi_2))]$,*

$$\sum_{k=1}^{K} b_k^r(\xi_{H,1,k}, \xi_{H,2,k}) \leq \mathcal{O}(dim_{\mathbb{T}}\sqrt{K \log(KB\rho_w) \log\left(\frac{K\left(1 + 2B\rho_w\right)}{\delta}\right)}) \tag{31}$$

*Proof.* Let $\mathcal{F}_k = \{f_r | f_r(x, y) = \sigma(r(x) - r(y)), r \in \mathcal{B}_k^r\}$, then we define $diam(\mathcal{F}_{\mathcal{B}_k^r}) = b_k^\sigma(x, y) = \max_{r_1, r_2 \in \mathcal{B}_k^r} \sigma(r_1(x) - r_1(y)) - \sigma(r_2(x) - r_2(y))$. According to Lemma. D.1, $\delta_k = \max_{1 \leq k \leq K} diam\left(\mathcal{F}_k|_{x_{1:K}}\right)$. Let $\alpha = 1/K$, $T = k$, and $d = \dim_\mathcal{E}(\mathcal{F}_\mathcal{R}, 1/K)$ According to lemma G.6,

$$\sum_{k=1}^K b_k^r(\xi_{H,1,k}, \xi_{H,2,k}) \leq \mathcal{O}(dim_\mathbb{T} \sqrt{K \log(KB\rho_w) \log\left(\frac{K(1 + 2B\rho_w)}{\delta}\right)}) \tag{32}$$

$\square$

**Lemma D.4.** *Transition estimation error of fixed state action pair.For any fixed $k$, with probability at least $1 - 2\delta$, for any $(s, a) \in \mathcal{S} \times \mathcal{A}$.*

$$\sum_{s' \in \mathcal{S}} \left|\hat{\mathbf{P}}_k(s' \mid s, a) - \mathbf{P}^\star(s' \mid s, a)\right| \leq \sqrt{\frac{2S \log\left(\frac{2KHSA}{\delta}\right)}{n_k(s, a)}}$$

*Proof.* The proof is same as Eq. (55) in Zanette and Brunskill [2019]. $\square$

**Lemma D.5.** *Transition estimation error of whole distribution.For any fixed $k$, with probability at least $1 - \delta$, for any $(s, a) \in \mathcal{S} \times \mathcal{A}$, excuted policy $\pi_1, \pi_2 \in \Pi_k$and any transition possibility kernel $\mathbf{P}_1, \mathbf{P}_2 \in \mathcal{B}_k^\mathbf{P}$*

$$\sum_{i=1}^2 \sum_{k=1}^K \max_{\mathbf{P}_1, \mathbf{P}_2 \in \mathcal{B}_k^\mathbf{P}} E_{\xi_i \sim \pi_i}(\sum_{h=1}^H |\mathbf{P}_1(s' \mid s_{i,k,h}, a_{i,k,h}) - \mathbf{P}_2(s' \mid s_{i,k,h}, a_{i,k,h})|) \tag{33}$$

$$\leq \mathcal{O}\left(S^2 A H^{3/2} \sqrt{K \log(K) log(K/\delta)}\right) \tag{34}$$

*Proof.* The proof is same as Lemma. A.5 in Chen et al. [2022]. $\square$

**Lemma D.6.** *For any iteration value $V : S \to R$, any two transition possibility kernel $\mathbf{P}, \hat{\mathbf{P}} : S \times S \times A \to$, and the risk aware object form:*

$$\Phi(V(s')) = \int_0^1 F_{V(s')}^\dagger(\xi) \cdot dG(\xi) \quad s' \sim \mathbf{P}(s, a) \tag{35}$$

$$\left|\Phi_{s' \sim \hat{\mathbf{P}}(\cdot | s, a)}(V(s')) - \Phi_{s' \sim \mathbf{P}(\cdot | s, a)}(V(s'))\right|$$
$$\leq L_G H \sum_{s' \in \mathcal{S}} \left|\hat{\mathbf{P}}(s' \mid s, a) - \mathbf{P}(s' \mid s, a)\right| \tag{36}$$

*Proof.* We firstly sort all successor states $s' \in \mathcal{S}$ by $V(s')$ in ascending order (from the left to the right) as $s_1', s_2' \dots s_S'$. And we assume that $V(s_{S+1}' = 1)$. Thus, according to the quantile function's definition,

$$\left|\Phi_{s' \sim \hat{\mathbf{P}}(\cdot | s, a)}(V(s')) - \Phi_{s' \sim \mathbf{P}(\cdot | s, a)}(V(s'))\right| \tag{37}$$

$$= |\int_0^1 F_{V\hat{\mathbf{P}}(s')}^\dagger(\xi) \cdot dG(\xi) - \int_0^1 F_{V\mathbf{P}(s')}^\dagger(\xi) \cdot dG(\xi)| \tag{38}$$

$$= |\int_0^1 G(F_{V\hat{\mathbf{P}}(s')}(\xi)) \cdot d\xi - \int_0^1 G(F_{V\mathbf{P}(s')}(\xi)) \cdot d\xi| \tag{39}$$

$$= \sum_{i=1}^S |V(s_{i+1}') - V(s_i')| \cdot |G(\sum_{j=1}^i \mathbf{P}(s_j' | (s, a))) - G(\sum_{j=1}^i \hat{\mathbf{P}}(s_j' | (s, a)))| \tag{40}$$

$$\leq \sum_{i=1}^S (V(s_{i+1}') - V(s_i')) \cdot L_G \sum_{j=1}^i |\mathbf{P}(s_j' | (s, a)) - \hat{\mathbf{P}}(s_j' | (s, a))| \tag{41}$$

$$\leq \sum_{i=1}^{S} (V(s'_{i+1}) - V(s'_i)) \cdot L_G \sum_{j=1}^{S} |\mathbf{P}(s'_j|(s,a)) - \hat{\mathbf{P}}(s'_j|(s,a))| \tag{42}$$

$$\leq L_G \sum_{j=1}^{S} |\mathbf{P}(s'_j|(s,a)) - \hat{\mathbf{P}}(s'_j|(s,a))| \sum_{i=1}^{S} (V(s'_{i+1}) - V(s'_i)) \tag{43}$$

$$\leq L_G \cdot H \sum_{j=1}^{S} |\mathbf{P}(s'_j|(s,a)) - \hat{\mathbf{P}}(s'_j|(s,a))| \tag{44}$$

$\square$

The second to fourth lines follow the discussion on the equivalent expression of the discontinuous distribution $\Phi$ in Lemma 5.1 of Bastani et al. [2022]. The fifth line is derived from the properties of Lipschitz functions and Assumption 3.5.

Since we use the emperical estimation $\hat{\mathbf{P}}_k$ of the transaction kernel $\mathbf{P}^\star$,

$$\hat{\mathbf{P}}_k = argmin_{\mathbf{P} \in \mathcal{P}} \sum_{i=1}^{2} \sum_{k'=1}^{k-1} \sum_{h=1}^{H} |\langle \mathbf{P}(s_{i,k',h}, a_{i,k',h}), \mathbb{I}(s_{i,k',h+1}) \rangle|^2. \tag{45}$$

**Lemma D.7.** *Concentration for V.*

*With probability at least $1 - 2\delta$, it holds that, for any $k \in [K]$, $(s,a) \in \mathcal{S} \times \mathcal{A}$, any transition possibility kernel $\mathbf{P}_1 \in \mathcal{B}_k^{\mathbf{P}}$ and function $V : \mathcal{S} \mapsto [0, H]$,*

$$\left| \Phi_{s' \sim \mathbf{P}_1(\cdot|s,a)}(V(s')) - \Phi_{s' \sim \mathbf{P}^\star(\cdot|s,a)}(V(s')) \right| \leq 2L_G \cdot H \sqrt{\frac{2S \log\left(\frac{2KHSA}{\delta}\right)}{n_k(s,a)}} \tag{46}$$

*Here $n_k(s,a)$ is the number of times $(s,a)$ was visited up to episode $k$.*

*Proof.* According to Lemma D.4 and recall the definition of the transition possibility confidence set, with probability at least $1 - 2\delta$,

$$\sum_{s' \in \mathcal{S}} |\mathbf{P}^\star(s' \mid s,a) - \mathbf{P}_1(s' \mid s,a)| \tag{47}$$

$$\leq \sum_{s' \in \mathcal{S}} \left| \mathbf{P}^\star(s' \mid s,a) - \hat{\mathbf{P}}_k(s' \mid s,a) + \hat{\mathbf{P}}_k(s' \mid s,a) - \mathbf{P}_1(s' \mid s,a) \right| \tag{48}$$

$$\tag{49}$$

$$\leq \sum_{s' \in \mathcal{S}} \left| \hat{\mathbf{P}}_k(s' \mid s,a) - \mathbf{P}^\star(s' \mid s,a) \right| + \sum_{s' \in \mathcal{S}} \left| \hat{\mathbf{P}}_k(s' \mid s,a) - \mathbf{P}_1(s' \mid s,a) \right| \tag{50}$$

$$\leq 2 \sqrt{\frac{2S \log\left(\frac{2KHSA}{\delta}\right)}{n_k(s,a)}} \tag{51}$$

Plugging Lemma .D.6, we obtain that with probability at least $1 - 2\delta$, for any $k \in [K]$, $(s,a) \in \mathcal{S} \times \mathcal{A}$ and function $V : \mathcal{S} \mapsto [0, H]$,

$$\left| \Phi_{s' \sim \hat{\mathbf{P}}^k(\cdot|s,a)}(V(s')) - \Phi_{s' \sim \mathbf{P}^\star(\cdot|s,a)}(V(s')) \right| \leq 2L_G \cdot H \sqrt{\frac{2S \log\left(\frac{2KHSA}{\delta}\right)}{n_k(s,a)}} \tag{52}$$

$\square$

Recall the definition of quantile function $\Phi$, $G$ is the quantile CDF weight. Given any target value $V : S \to R$, we use $\beta_{\mathbf{P}^\star}^{G,V}(s' \mid s, a)$ denotes the conditional probability of transitioning to $s'$ from $(s, a)$, conditioning on transitioning to the quantile distribution, and it holds that

$$\beta_{\mathbf{P}^\star}^{G,V}(s'_i \mid s, a) = G(\sum_{j=1}^{i+1} \mathbf{P}^\star(s'_j \mid (s, a))) - G(\sum_{j=1}^{i} \mathbf{P}^\star(s'_j \mid (s, a))) \tag{53}$$

**Lemma D.8.** *Quantile Reward Gap due to Value Function Shift.* *For any $(s, a) \in \mathcal{S} \times \mathcal{A}$, distribution $\mathbf{P}$, and functions $V, V' : \mathcal{S} \mapsto [0, H]$, for any $s' \in \mathcal{S}$,*

$$\Phi_{s' \sim \mathbf{P}(\cdot \mid s, a)}(V'(s')) - \Phi_{s' \sim \mathbf{P}(\cdot \mid s, a)}(V(s')) \leq \beta_{\mathbf{P}}^{G,V}(\cdot \mid s, a)^\top |V' - V| \tag{54}$$

This Lemma's proof is similar to Lemma 11 in Du et al. [2022].

**Lemma D.9.** *For any $(s, a) \in \mathcal{S} \times \mathcal{A}$, distribution $\mathbf{P}$, and functions $V, V' : \mathcal{S} \mapsto [0, H]$, for any $s' \in \mathcal{S}$,*

$$\Phi_{s' \sim \mathbf{P}(\cdot \mid s, a)}(V'(s')) - \Phi_{s' \sim \mathbf{P}(\cdot \mid s, a)}(V(s')) \leq L_G \mathbf{P}(\cdot \mid s, a)^\top |V' - V| \tag{55}$$

*Proof.* This Lemma comes from Lemma .D.8 and

$$\beta^{G,V}(s'_i \mid s, a) = G(\sum_{j=1}^{i+1} \mathbf{P}^\star(s'_j \mid (s, a))) - G(\sum_{j=1}^{i} \mathbf{P}^\star(s'_j \mid (s, a))) \tag{56}$$

$$\leq L_G \mathbf{P}(\cdot \mid s, a) \tag{57}$$

$\square$

For any $k > 0, h \in [H]$ and $(s, a) \in \mathcal{S} \times \mathcal{A}$, let $p_{kh}(s, a)$ denote the probability of visiting $(s, a)$ at step $h$ of episode $k$. Then, it holds that for any $k > 0, h \in [H]$ and $(s, a) \in \mathcal{S} \times \mathcal{A}, p_{kh}(s, a) \in [0, 1]$ and $\sum_{(s,a) \in \mathcal{S} \times \mathcal{A}} p_{kh}(s, a) = 1$

**Lemma D.10.** *(Concentration of state-action visitation).* *It holds that*

$$\Pr\left[ n_k(s, a) \geq \frac{1}{2} \sum_{k'=1}^{k-1} \sum_{h=1}^{H} p_{k'h}(s, a) - H \log\left(\frac{HSA}{\delta}\right), \forall k > 0, \forall (s, a) \in \mathcal{S} \times \mathcal{A} \right] \geq 1 - \delta$$

This Lemma is a direct application of Lemma G.9 and same as Du et al. [2022].

**Lemma D.11.** *(Concentration of trajectory visitation).* *It holds that*

$$\Pr\left[ n_{dim,k}(d) \geq \frac{1}{2} \sum_{k'=1}^{k-1} p_{dim,k'}(\xi_H) - \log\left(\frac{dim_{\mathbf{T}}}{\delta}\right), \forall d \in [0, \ldots, dim_{\mathbb{T}}] \right] \geq 1 - \delta$$

*Proof.* This Lemma a direct application of G.9. For any dimension $d \in [0, \ldots, dim_{\mathbb{T}}]$, it holds that

$$\Pr\left[ n_{dim,k}(d) \geq \frac{1}{2} \sum_{k'=1}^{k-1} p_{dim,k'}(\xi_H) - \log\left(\frac{1}{\delta}\right) \right] \geq 1 - \delta$$

Since $d \in [0, \ldots, dim_{\mathbb{T}}]$, Therefore,

$$\Pr\left[ n_{dim,k}(d) \geq \frac{1}{2} \sum_{k'=1}^{k-1} p_{dim,k'}(\xi_H) - \log\left(\frac{dim_{\mathbf{T}}}{\delta}\right), d \in [0, \ldots, dim_{\mathbb{T}}] \right] \geq 1 - \delta$$

$\square$

**Definition of sufficient state-action visitations.** Following Zanette and Brunskill [2019], for any episode $k > 0$, we define the set of state-action pairs which have sufficient visitations in expectation as follows.

$$\mathcal{L}_k := \left\{ (s,a) \in \mathcal{S} \times \mathcal{A} : \frac{1}{4} \sum_{k'=1}^{k-1} \sum_{h=1}^{H} p_{k'h}(s,a) \geq H \log\left(\frac{HSA}{\delta}\right) + H \right\}$$

**Definition of sufficient trajectory visitations.** Following Zanette and Brunskill [2019], for any episode $k > 0$, we define the set of trajrctory dimension which have sufficient visitations in expectation as follows.

$$\mathcal{L}_{dim,k} := \left\{ d \in [0,\dots,dim_{\mathbb{T}}] : \frac{1}{4} \sum_{k'=1}^{k-1} p_{dim,k'}(d) \geq \log\left(\frac{dim_{\mathbb{T}}}{\delta}\right) + 1 \right\}$$

We use $n_{dim,k}(d)$ to denote the number of $\phi_d(\xi_H) \neq 0$ among $1 \sim k$ episode's trajectory.

**Lemma D.12.** *(Standard state action visitation ratio). For any $K > 0$, we have*

$$\sqrt{\sum_{k=1}^{K} \sum_{h=1}^{H} \sum_{(s,a) \in \mathcal{L}_k} \frac{p_{kh}(s,a)}{n_k(s,a)}} \leq 2\sqrt{SA \log\left(\frac{KHSA}{\delta}\right)}.$$

This proof is the same as that of Lemma 13 in Zanette and Brunskill [2019].

**Lemma D.13.** *(Standard trajectory visitation ratio). For any $K > 0$, we have*

$$\sqrt{\sum_{k=1}^{K} \sum_{d=1}^{dim_{\mathbb{T}}} \frac{p_{dim,k}(d)}{n_{dim,k}(d)})} \leq 2\sqrt{dim_{\mathbb{T}} \log\left(\frac{K dim_{\mathbb{T}}}{\delta}\right)}.$$

This proof is the same as that of Lemma 13 in Zanette and Brunskill [2019].

**Lemma D.14.** *(Standard Invisitation Ratio). For any $K > 0$, we have*

$$\sum_{k=1}^{K} \sum_{h=1}^{H} \sum_{(s,a) \notin \mathcal{L}_k} p_{kh}(s,a) < \frac{1}{\min_{\pi,h,s:p_{\pi,h}(s)>0} p_{\pi,h}(s)} \cdot \left(4H \log\left(\frac{HSA}{\delta}\right) + 5H\right) \qquad (58)$$

This proof is the same as that of Lemma 10 in Du et al. [2022].

**Lemma D.15.** *(Standard trajectory Invisitation Ratio). For any $K > 0$, we have*

$$\sum_{k=1}^{K} \sum_{d \notin \mathcal{L}_{d,k}} p_{dim,k}(d) < \min_{\pi,d:p_{dim,\pi}(d)>0} p_{dim,\pi}(d) \cdot \left(4 \log\left(\frac{dim_{\mathbb{T}}}{\delta}\right) + 5\right)$$

This proof is the same as that of Lemma 10 in Du et al. [2022].

**Lemma D.16.** *For any functions $V : \mathcal{S} \mapsto \mathbb{R}, k > 0, h \in [H]$ and $(s,a) \in \mathcal{S} \times \mathcal{A}$ such that $p_{kh}(s,a) > 0$,*

$$\frac{p_{kh}^{G,V}(s,a)}{p_{kh}(s,a)} \leq \frac{1}{\min_{\pi,h,(s,a):w_{\pi,h}(s,a)>0} w_{\pi,h}(s,a)},$$

*where $p_{kh}^{G,V}(s,a)$ denotes the conditional probability of visiting $(s,a)$ at step $h$ of episode $k$, conditioning on transitioning distortion by the quantion function $G$ which works on at each step $h' = 1,\dots,h-1$.*

*Proof.* Since $p_{kh}^{G,V}(s,a)$ is the conditional probability of visiting $(s,a)$, we have $p_{kh}^{Ga,V}(s,a) \in [0,1]$. Since $p_{kh}(s,a)$ is the probability of visiting $(s,a)$ at step $h$ under policy $\pi^k$ and $\min_{\pi,h,(s,a):w_{\pi,h}(s,a)>0} w_{\pi,h}(s,a)$ is the minimum probability of visiting any reachable $(s,a)$ at any step $h$ over all policies $\pi$, we have

$$p_{kh}(s,a) \geq \min_{\pi,h,(s,a):w_{\pi,h}(s,a)>0} w_{\pi,h}(s,a).$$

Hence, we have

$$\frac{p_{kh}^{G,\alpha,V}(s,a)}{p_{kh}(s,a)} \leq \frac{1}{\min_{\pi,h,(s,a):w_{\pi,h}(s,a)>0} w_{\pi,h}(s,a)}.$$

$\square$

**Lemma D.17.** *For any functions* $V : \mathcal{S} \mapsto \mathbb{R}, k > 0, h \in [H]$,

$$\frac{p_{dim,k}^{G,V}(d)}{p_{dim,k}(d)} \leq \frac{1}{\min p_\pi(d)},$$

*where* $p_{dim,k}^{G,V}(d)$ *denotes the conditional probability of* $\phi_d(\xi_H) \neq 0$ *of episode* $k$*, conditioning on transitioning distortion by the quantion function* $G$ *which always works on at each step* $h = 1,\ldots,H$.

The proof is similar to Lemma .D.16.

## D.2 Proof of Algorithm 1's regret upper bound

**Lemma D.18.** *For a probability of* $1 - 2\delta$*, it holds that* $\pi^\star \in \Pi_k$*, which is calculated as follows:*

$$\Pi_k = \{\pi \mid \max_{r_\xi \in \mathcal{B}_k^r, \mathbf{P} \in \mathcal{B}_k^{\mathbf{P}}} (\tilde{V}_{1,r_\xi,\mathbf{P}}^\pi(\xi_h) - \tilde{V}_{1,r_\xi,\mathbf{P}}^{\pi_0}(\xi_h)) \geq 0, \forall \pi_0\} \tag{59}$$

*Proof.* It equals to for any policy $\pi_0 \in \Pi$,

$$\max_{r_\xi \in \mathcal{B}_k^r, \mathbf{P} \in \mathcal{B}_k^{\mathbf{P}}} (\tilde{V}_{1,r_\xi,\mathbf{P}}^{\pi^\star}(\xi_h) - \tilde{V}_{1,r_\xi,\mathbf{P}}^{\pi_0}(\xi_h)) \geq 0 \tag{60}$$

According to Lemma D.1 and Lemma D.4, with at least possibility $1 - 2\delta$, it holds that:

$$r_\xi^\star \in \mathcal{B}_k^r, \mathbf{P}^\star \in \mathcal{B}_k^{\mathbf{P}} \tag{61}$$

Thus,

$$\max_{r_\xi \in \mathcal{B}_k^r, \mathbf{P} \in \mathcal{B}_k^{\mathbf{P}}} (\tilde{V}_{1,r_\xi,\mathbf{P}}^{\pi^\star}(\xi_h) - \tilde{V}_{1,r_\xi,\mathbf{P}}^{\pi_0}(\xi_h)) \tag{62}$$

$$\geq \tilde{V}_{1,r_\xi,\mathbf{P}}^{\pi^\star}(\xi_h) - \tilde{V}_{1,r_\xi,\mathbf{P}}^{\pi_0}(\xi_h) \geq 0 \tag{63}$$

Where the last equation comes from the definition of optimal policy $\pi^\star$ (Eq. 7). $\square$

**Lemma D.19.** *Given a positive constant* $\delta \in (0,1]$*, with probability at least* $1 - 4K\delta$*, we have the following inequality holds:*

$$\text{Reg}_{nested}(K) \tag{64}$$

$$\leqslant \max_{r_1 \in \mathcal{B}_k^r, \mathbf{P}_1 \in \mathcal{B}_k^{\mathbf{P}}} \{\tilde{V}_{1,r^\star,\mathbf{P}^\star}^{\pi^\star}(s_{k,1}) - \tilde{V}_{1,r^\star,\mathbf{P}^\star}^{\pi_1}(s_{k,1}) - (\tilde{V}_{1,r_1,\mathbf{P}_1}^{\pi^\star}(s_{k,1}) - \tilde{V}_{1,r_1,\mathbf{P}_1}^{\pi_1}(s_{k,1}))\} \tag{65}$$

$$+ \max_{r_2 \in \mathcal{B}_k^r, \mathbf{P}_2 \in \mathcal{B}_k^{\mathbf{P}}} \{\tilde{V}_{1,r^\star,\mathbf{P}^\star}^{\pi^\star}(s_{k,1}) - \tilde{V}_{1,r^\star,\mathbf{P}^\star}^{\pi_2}(s_{k,1}) - (\tilde{V}_{1,r_2,\mathbf{P}_2}^{\pi^\star}(s_{k,1}) - \tilde{V}_{1,r_2,\mathbf{P}_2}^{\pi_2}(s_{k,1}))\} \tag{66}$$

*Proof.*

$$\text{Reg}(K) = \sum_{k=1}^K \left(\tilde{V}_{1,r^\star,\mathbf{P}^\star}^{\pi^\star}(s_{k,1}) - \tilde{V}_{1,r^\star,\mathbf{P}^\star}^{\pi_1}(s_{k,1}) + \tilde{V}_{1,r^\star,\mathbf{P}^\star}^{\pi^\star}(s_{k,1}) - \tilde{V}_{1,r^\star,\mathbf{P}^\star}^{\pi_2}(s_{k,1})\right) \tag{67}$$

$$= \sum_{k=1}^K \max_{r_1 \in \mathcal{B}_k^r, \mathbf{P}_1 \in \mathcal{B}_k^{\mathbf{P}}} (\tilde{V}_{1,r_1,\mathbf{P}_1}^{\pi^\star}(s_{k,1}) - \tilde{V}_{1,r_1,\mathbf{P}_1}^{\pi_1}(s_{k,1})) \tag{68}$$

$$+ \tilde{V}_{1,r^\star,\mathbf{P}^\star}^{\pi^\star}(s_{k,1}) - \tilde{V}_{1,r^\star,\mathbf{P}^\star}^{\pi_1}(s_{k,1}) - \max_{r_1 \in \mathcal{B}_k^r, \mathbf{P}_1 \in \mathcal{B}_k^{\mathbf{P}}} (\tilde{V}_{1,r_1,\mathbf{P}_1}^{\pi^\star}(s_{k,1}) - \tilde{V}_{1,r_1,\mathbf{P}_1}^{\pi_1}(s_{k,1}))$$

$$\tag{69}$$

$$+\sum_{k=1}^{K}\max_{r_2\in\mathcal{B}_k^r,\mathbf{P}_2\in\mathcal{B}_k^{\mathbf{P}}}(\tilde{V}_{1,r_2,\mathbf{P}_2}^{\pi^\star}(s_{k,1})-\tilde{V}_{1,r_2,\mathbf{P}_2}^{\pi_2}(s_{k,1}))\tag{70}$$

$$+\tilde{V}_{1,r^\star,\mathbf{P}^\star}^{\pi^\star}(s_{k,1})-\tilde{V}_{1,r^\star,\mathbf{P}^\star}^{\pi_2}(s_{k,1})-\max_{r_2\in\mathcal{B}_k^r,\mathbf{P}_2\in\mathcal{B}_k^{\mathbf{P}}}(\tilde{V}_{1,r_2,\mathbf{P}_2}^{\pi^\star}(s_{k,1})-\tilde{V}_{1,r_2,\mathbf{P}_2}^{\pi_2}(s_{k,1}))\tag{71}$$

$$\overset{a}{\leq}\tilde{V}_{1,r^\star,\mathbf{P}^\star}^{\pi^\star}(s_{k,1})-\tilde{V}_{1,r^\star,\mathbf{P}^\star}^{\pi_1}(s_{k,1})-\max_{r_1\in\mathcal{B}_k^r,\mathbf{P}_1\in\mathcal{B}_k^{\mathbf{P}}}(\tilde{V}_{1,r_1,\mathbf{P}_1}^{\pi^\star}(s_{k,1})-\tilde{V}_{1,r_1,\mathbf{P}_1}^{\pi_1}(s_{k,1}))\tag{72}$$

$$+\tilde{V}_{1,r^\star,\mathbf{P}^\star}^{\pi^\star}(s_{k,1})-\tilde{V}_{1,r^\star,\mathbf{P}^\star}^{\pi_2}(s_{k,1})-\max_{r_2\in\mathcal{B}_k^r,\mathbf{P}_2\in\mathcal{B}_k^{\mathbf{P}}}(\tilde{V}_{1,r_2,\mathbf{P}_2}^{\pi^\star}(s_{k,1})-\tilde{V}_{1,r_2,\mathbf{P}_2}^{\pi_2}(s_{k,1}))\tag{73}$$

$$\overset{b}{\leq}\max_{r_1\in\mathcal{B}_k^r,\mathbf{P}_1\in\mathcal{B}_k^{\mathbf{P}}}\{\tilde{V}_{1,r^\star,\mathbf{P}^\star}^{\pi^\star}(s_{k,1})-\tilde{V}_{1,r^\star,\mathbf{P}^\star}^{\pi_1}(s_{k,1})-(\tilde{V}_{1,r_1,\mathbf{P}_1}^{\pi^\star}(s_{k,1})-\tilde{V}_{1,r_1,\mathbf{P}_1}^{\pi_1}(s_{k,1}))\}\tag{74}$$

$$+\max_{r_2\in\mathcal{B}_k^r,\mathbf{P}_2\in\mathcal{B}_k^{\mathbf{P}}}\{\tilde{V}_{1,r^\star,\mathbf{P}^\star}^{\pi^\star}(s_{k,1})-\tilde{V}_{1,r^\star,\mathbf{P}^\star}^{\pi_2}(s_{k,1})-(\tilde{V}_{1,r_2,\mathbf{P}_2}^{\pi^\star}(s_{k,1})-\tilde{V}_{1,r_2,\mathbf{P}_2}^{\pi_2}(s_{k,1}))\}\tag{75}$$

Where (a) comes from the definition of the optimal policy confidence set (Lemma D.18) when $\pi^\star\in\Pi_k$ at each episode , (b) derives from the characters of max value. $\square$

### D.3 Nested Regret

**Lemma D.20.** *Nested regret upper bound. Given a positive constant $\delta\in(0,1]$, with probability at least $1-\delta$, we have the following inequality holds for every $k\in[K]$.*

$$\mathrm{Reg}_{nested}(K)$$

$$\leq\mathcal{O}\left(L_GH^{\frac{3}{2}}\sqrt{K}SA\log\left(\frac{KHSA}{\delta}\right)\cdot\frac{1}{\sqrt{\min_{\pi,h,(s,a):p_{\pi,h}(s,a)>0}w_{\pi,h}(s,a)}}\right)$$

$$+\mathcal{O}\left(\frac{B}{\kappa b}dim_{\mathbb{T}}\sqrt{\log\left(\frac{Kdim_{\mathbb{T}}}{\delta}\right)\log\left(\frac{K(1+2B\rho_w)}{\delta}\right)}\frac{1}{min\omega_\pi(d)}\right)\tag{76}$$

*Proof.* We use $V_{1,r^\star,p}^{\pi,h}(s_{k,1})$ to denote that the first $h$ steps in the trajectory $\xi$ is sampled using policy $\pi$ from the MDP with transition $\widehat{\mathbf{P}}$, and the state-action pairs from step $h+1$ up until the last step is sampled using policy $\pi$ from the MDP with the true transition kernal $\mathbf{P}^\star$. Therefore,

Here (a) is due to Lemma D.7 and D.8. (b) comes from that $p_{kh}^{CVaR,\alpha,V^{\pi^k}}(s,a)$ is defined as the probability of visiting $(s,a)$ at step $h$ of episode $k$ under the conditional transition probability $\beta^{\alpha,V_{h'+1}^{\pi^k}}(\cdot\mid\cdot,\cdot)$ for each step $h'=1,\ldots,h-1$.

Firstly, we analyze the term $I_1$ and $I_5$. Recall that for any policy $\pi,h\in[H]$ and $(s,a)\in\mathcal{S}\times\mathcal{A}$, $w_{\pi,h}(s,a)$ and $w_{\pi,h}(s)$ denote th probabilities of visiting $(s,a)$ and $s$ at step $h$ under policy $\pi$, respectively. Thus, we have:

$$\sum_{k=1}^{K}\sum_{h=1}^{H}\sum_{(s,a)\in\mathcal{L}_k}p_{kh}^{G,\alpha,V^{\pi^k}}(s,a)L_G\cdot H\sqrt{\frac{2SA\log(\frac{2KHSA}{\delta})}{n_k(s,a)}}\tag{77}$$

$$\overset{(a)}{\leq}L_GH\sqrt{2SA\log(\frac{2KHSA}{\delta})}\sqrt{\sum_{k=1}^{K}\sum_{h=1}^{H}\sum_{(s,a)\in\mathcal{L}_k}\frac{p_{kh}^{G,\alpha,V^{\pi^k}}(s,a)}{n_k(s,a)}}\cdot\sqrt{\sum_{k=1}^{K}\sum_{h=1}^{H}\sum_{(s,a)\in\mathcal{L}_k}p_{kh}^{G,\alpha,V^{\pi^k}}(s,a)}\tag{78}$$

$$= L_G H \sqrt{2SA \log(\frac{2KHSA}{\delta})} \sqrt{\sum_{k=1}^{K} \sum_{h=1}^{H} \sum_{(s,a) \in \mathcal{L}_k} \frac{p_{kh}^{G,\alpha,V^{\pi^k}}(s,a)}{n_k(s,a)} \cdot \nVdash \{p_{kh}(s,a) \neq 0\}} \cdot \sqrt{KH} \tag{79}$$

$$= L_G H \sqrt{2SAKH \log(\frac{2KHSA}{\delta})} \sqrt{\sum_{k=1}^{K} \sum_{h=1}^{H} \sum_{(s,a) \in \mathcal{L}_k} \frac{p_{kh}^{G,\alpha,V^{\pi^k}}(s,a)}{p_{kh}(s,a)} \cdot \frac{p_{kh}(s,a)}{n_k(s,a)} \cdot \nVdash \{p_{kh}(s,a) \neq 0\}} \tag{80}$$

$$\overset{(b)}{\leq} L_G H \sqrt{2SAKH \log(\frac{2KHSA}{\delta})} \sqrt{\frac{1}{\min_{\pi,h,(s,a):w_{\pi,h}(s,a)>0} w_{\pi,h}(s,a)} \sum_{k=1}^{K} \sum_{h=1}^{H} \sum_{(s,a) \in \mathcal{L}_k} \frac{p_{kh}(s,a)}{n_k(s,a)}} \tag{81}$$

$$\overset{(c)}{\leq} L_G H \sqrt{2SAKH \log(\frac{2KHSA}{\delta})} \cdot \sqrt{\frac{2S \log\left(\frac{2KHSA}{\delta}\right)}{n_k(s,a)}} \cdot \frac{1}{\sqrt{\min_{\pi,h,(s,a):p_{\pi,h}(s,a)>0} w_{\pi,h}(s,a)}} \tag{82}$$

$$\leq 2 L_G H^{\frac{3}{2}} \sqrt{K} SA \log\left(\frac{2KHSA}{\delta}\right) \cdot \frac{1}{\sqrt{\min_{\pi,h,(s,a):p_{\pi,h}(s,a)>0} w_{\pi,h}(s,a)}} \tag{83}$$

Here (a) is due to Cauchy-Schwartz inequality and the fact that for any $k > 0$ and $h \in [H]$, $\sum_{(s,a) \in \mathcal{S} \times \mathcal{A}} p_{kh}^{\text{CVaR},\alpha,V^{\pi^k}}(s,a) = 1$. (b) comes from Lemma D.16. (c) uses Lemma D.12 and the fact that for any deterministic policy $\pi, h \in [H]$ and $(s,a) \in \mathcal{S} \times \mathcal{A}$, we have either $w_{\pi,h}(s,a) = w_{\pi,h}(s)$ or $w_{\pi,h}(s,a) = 0$, and thus $\min_{\pi,h,(s,a):w_{\pi,h}(s,a)>0} w_{\pi,h}(s,a) = \min_{\pi,h,s:w_{\pi,h}(s)>0} w_{\pi,h}(s)$.

For the term $I_2$ and $I_6$, using Lemma D.14, we have:

$$\sum_{k=1}^{K} \sum_{h=1}^{H} \left( \sum_{(s,a) \notin \mathcal{L}_k} p_{k,h}^{G,V_{r^\star}^\pi,\mathbf{P}^\star}(s,a) H \right) \tag{84}$$

$$\leq \frac{1}{\min_{\pi,h,s:w_{\pi,h}(s)>0} w_{\pi,h}(s)} \left( 8SAH^2 \log\left(\frac{HSA}{\delta}\right) + 10SAH^2 \right) \tag{85}$$

For the term $I_3$ and $I_7$, we have:

$$\sum_{k=1}^{K} \sum_{d \in \mathcal{L}_{dim,k}} p_{dim,k}^{G,V_{r^\star}^{\pi_k},\mathbf{P}^\star}(d) \left| (w_{r^\star} - w_{\hat{r}_k}) B \right| \tag{86}$$

$$\overset{a}{\leq} \sqrt{\sum_{k=1}^{K} \sum_{d \in \mathcal{L}_{dim,k}} p_{dim,k}^{G,V_{r^\star}^{\pi_k},\mathbf{P}^\star}(d)} \sqrt{\frac{\sum_{k=1}^{K} \sum_{d \in \mathcal{L}_{dim,k}} p_{dim,k}^{G,V_{r^\star}^{\pi_k},\mathbf{P}^\star}(d)}{n_{dim,K}(d)}} \sqrt{n_{dim,k}(d) \sum_{k=1}^{K} \sum_{d \in \mathcal{L}_{dim,k}} |(w_{r^\star} - w_{\hat{r}_k}) B|} \tag{87}$$

$$\overset{b}{\leq} \frac{B}{\kappa b} \sqrt{dim_{\mathbb{T}} \log\left(\frac{K(1+2B\rho_w)}{\delta}\right)} \sqrt{\frac{\sum_{k=1}^{K} \sum_{d \in \mathcal{L}_{dim,k}} p_{dim,k}^{G,V_{r^\star}^{\pi_k},\mathbf{P}^\star}(d)}{n_{dim,K}(d)}} \tag{88}$$

$$\leq \frac{B}{\kappa b} \sqrt{dim_{\mathbb{T}} \log\left(\frac{K(1+2B\rho_w)}{\delta}\right)} \sqrt{\frac{\sum_{k=1}^{K} \sum_{d \in \mathcal{L}_{dim,k}} p_{dim,k}^{G,V_{r^\star}^{\pi_k},\mathbf{P}^\star}(d)}{p_{dim,K}^{\pi_k}(d)} \cdot \frac{p_{dim,K}^{\pi_k}(d)}{n_{dim,K}(d)} \mathbb{I}(p_{dim,K}^{\pi_k}(d) \neq 0)} \tag{89}$$

$$\overset{c}{\leq} \frac{2B}{\kappa b} dim_{\mathbb{T}} \sqrt{\log\left(\frac{K dim_{\mathbb{T}}}{\delta}\right) \log\left(\frac{K(1+2B\rho_w)}{\delta}\right)} \sqrt{\frac{\sum_{k=1}^{K} \sum_{d \in \mathcal{L}_{dim,k}} p_{dim,k}^{G,V_{r^\star}^{\pi_k},\mathbf{P}^\star}(d)}{p_{dim,K}^{\pi_k}(d)} \cdot \mathbb{I}(p_{dim,K}^{\pi_k}(d) \neq 0)} \tag{90}$$

$$\overset{d}{\leq} \frac{2B}{\kappa b} dim_{\mathbb{T}} \sqrt{\log\left(\frac{K dim_{\mathbb{T}}}{\delta}\right) \log\left(\frac{K\left(1+2B\rho_w\right)}{\delta}\right)} \frac{1}{min\omega_\pi(d)} \tag{91}$$

Here (a) is due to Cauchy-Schwartz inequality. (b) comes from Lemma D.17. (c) uses Lemma D.15. For the term $I_4$ and $I_8$, using Lemma D.15, we have:

$$\sum_{k=1}^{K}\left(\sum_{(d\notin\mathcal{L}_{dim,k})} p_{dim,k}^{G,V_{r^\star,\mathbf{P}^\star}^{\pi_k}}(d)B\right) \tag{92}$$

$$\leq \min_{\pi,d:p_{\pi,dim}(d)>0} p_{\pi,dim}(d)\cdot\left(4\log\left(\frac{dim_{\mathbb{T}}}{\delta}\right)+5\right) \tag{93}$$

Then, summing all the term, we have:

$$\text{Reg}_{\text{nested}}(K)$$

$$\leq\mathcal{O}\left(L_G H^{\frac{3}{2}}\sqrt{K}SA\log\left(\frac{KHSA}{\delta}\right)\cdot\frac{1}{\sqrt{\min_{\pi,h,(s,a):p_{\pi,h}(s,a)>0} w_{\pi,h}(s,a)}}\right)$$

$$+\mathcal{O}\left(\frac{B}{\kappa b} dim_{\mathbb{T}}\sqrt{\log\left(\frac{K dim_{\mathbb{T}}}{\delta}\right)\log\left(\frac{K\left(1+2B\rho_w\right)}{\delta}\right)}\frac{1}{\min_{\pi,d}\omega_{dim,\pi}(d)}\right) \tag{94}$$

$\square$

## D.4   Static Regret

**Lemma D.21.** ***Static regret upper bound.*** *Given a positive constant $\delta\in(0,1]$, with probability at least $1-\delta$, we have the following inequality holds for every $k\in[K]$.*

$$\text{Reg}_{static}(K)$$

$$\leq\mathcal{O}\left(L_G S^2 A H^{\frac{3}{2}}\sqrt{K}log\left(K/\delta\right)\right)+\quad\mathcal{O}\left(L_G dim_{\mathbb{T}}\sqrt{K\log(KB\rho_w)\log\left(\frac{K\left(1+2B\rho_w\right)}{\delta}\right)}\right) \tag{95}$$

*Proof.* We use $V_{1,r^\star,p}^{\pi,h}\left(s_{k,1}\right)$ to denote that the first $h$ steps in the trajectory $\xi$ is sampled using policy $\pi$ from the MDP with transition $\widehat{\mathbf{P}}$, and the state-action pairs from step $h+1$ up until the last step is sampled using policy $\pi$ from the MDP with the true transition kernal $\mathbf{P}^\star$. Therefore,

$$\text{Reg}_{static}(K)= \tag{96}$$

$$\tilde{V}_{1,r^\star,\mathbf{P}^\star}^{\pi^\star}\left(s_{k,1}\right)-\tilde{V}_{1,r^\star,\mathbf{P}^\star}^{\pi_1}\left(s_{k,1}\right)-\left(\tilde{V}_{1,r_1,\mathbf{P}_1}^{\pi^\star}\left(s_{k,1}\right)-\tilde{V}_{1,r_1,\mathbf{P}_1}^{\pi_1}\left(s_{k,1}\right)\right) \tag{97}$$

$$\leq\tilde{V}_{1,r^\star,\mathbf{P}^\star}^{\pi^\star}\left(s_{k,1}\right)-\tilde{V}_{1,r^\star,\mathbf{P}^\star}^{\pi^\star}\left(s_{k,1}\right)-\left(\tilde{V}_{1,r^\star,\mathbf{P}_1}^{\pi^\star}\left(s_{k,1}\right)-\tilde{V}_{1,r^\star,\mathbf{P}_1}^{\pi^\star}\left(s_{k,1}\right)\right) \tag{98}$$

$$+\tilde{V}_{1,r^\star,\mathbf{P}_1}^{\pi^\star}\left(s_{k,1}\right)-\tilde{V}_{1,r^\star,\mathbf{P}_1}^{\pi^\star}\left(s_{k,1}\right)-\left(\tilde{V}_{1,r_1,\mathbf{P}_1}^{\pi^\star}\left(s_{k,1}\right)-\tilde{V}_{1,r_1,\mathbf{P}_1}^{\pi_1}\left(s_{k,1}\right)\right) \tag{99}$$

$$\leq\sum_{k=1}^{K} E_{\xi_1\sim\pi^\star,\xi_2\sim\pi_1}\left(\max_{\mathbf{P}_1,\mathbf{P}_2\in\mathcal{B}_k^\mathbf{P}}\left(\sum_{i=1}^{2}\sum_{h=1}^{H}|\mathbf{P}_1\left(s'\mid s_{i,k,h},a_{i,k,h}\right)-\mathbf{P}_2\left(s'\mid s_{i,k,h},a_{i,k,h}\right)|\right)\right) \tag{100}$$

$$+\sum_{k=1}^{K} E_{\xi_1\sim\pi^\star,\xi_2\sim\pi_1}\left(\max_{r_1,r_2\in\mathcal{B}_k^r}\left(\sum_{i=1}^{2}\sum_{h=1}^{H}\langle\mathbf{P}^\star\left(\cdot\mid s_{i,k,h},a_{i,k,h}\right),\tilde{V}^{r_1}\left(\cdot\mid s_{i,k,h},a_{i,k,h}\right)-\tilde{V}^{r_2}\left(\cdot\mid s_{i,k,h},a_{i,k,h}\right)\rangle\right)\right)$$

$$\tag{101}$$

$$\leq\mathcal{O}\left(L_G S^2 A H^{\frac{3}{2}}\sqrt{K}log\left(K/\delta\right)\right)+\mathcal{O}\left(L_G dim_{\mathbb{T}}\sqrt{K\log(KB\rho_w)\log\left(\frac{K\left(1+2B\rho_w\right)}{\delta}\right)}\right)$$

$$\tag{102}$$

$\square$

# E    Lower bound of the regret.

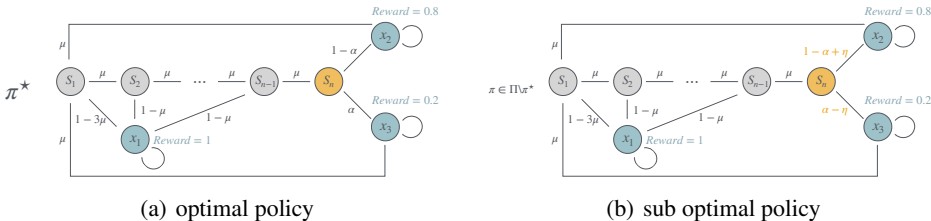

(a) optimal policy                           (b) sub optimal policy

Figure 5: Hard to learn case 1

## E.1    Regret Lower Bound of Nested Reward

**Lemma E.1.** *Nested Regret Lower Bound. There exists an instance of Nested CVaR RL-RM, where the regret of any algorithm is at least:*

$$\text{Regret}(K) \geq \mathcal{O}\left(\min\left\{B\rho_w\sqrt{\frac{AK}{\min_{\pi,h,s:p_{\pi,h}(s)>0} w_{\pi,h}(s,a)}}, B\sqrt{\frac{AK}{\min_{\pi,d} \omega_{dim,\pi}(d)}}\right\}\right) \quad (103)$$

*Proof.* **Hard to learn case 1.** Consider the instance shown in Figure 5. The state space is defined as $\mathcal{S} = \{s_1, s_2, \ldots, s_n, x_1, x_2, x_3\}$, where $s_1$ is the initial state, and $n = S - 3 < S$. We define the trajectory reward functions as follows: For any $\xi_H \in \mathcal{Z}_H$, $\phi(\xi_H) = (\mathbb{I}(S_H(\xi_H) = x_1)B, \mathbb{I}(S_H(\xi_H) = x_2)B, \mathbb{I}(S_H(\xi_H) = x_3)B)$, $w_r = (1\rho, 0.8\rho, 0.2\rho)$

The transition distributions are as follows. Let $\mu$ be a parameter which satisfies that $0 < \alpha < \mu < \frac{1}{3}$. For any $a \in \mathcal{A}, p(s_2 \mid s_1, a) = \mu, p(x_1 \mid s_1, a) = 1 - 3\mu, p(x_2 \mid s_1, a) = \mu$ and $p(x_3 \mid s_1, a) = \mu$. For any $i \in \{2, \ldots, n-1\}$ and $a \in \mathcal{A}, p(s_{i+1} \mid s_i, a) = \mu$ and $p(x_1 \mid s_i, a) = 1 - \mu. x_1, x_2$ and $x_3$ are absorbing states, i.e., for any $a \in \mathcal{A}, p(x_1 \mid x_1, a) = 1, p(x_2 \mid x_2, a) = 1$ and $p(x_3 \mid x_3, a) = 1$. $p(x_2 \mid s_n, a_J) = 1 - \alpha + \eta$ and $p(x_3 \mid s_n, a_J) = \alpha - \eta$, where $\eta$ is a parameter which satisfies $0 < \eta < \alpha$ and will be chosen later. For any suboptimal action $a \in \mathcal{A}\backslash\{a_J\}, p(x_2 \mid s_n, a) = 1 - \alpha$ and $p(x_3 \mid s_n, a) = \alpha$ For any $a_j \in \mathcal{A}$, let $\mathbb{E}_j[\cdot]$ and $\text{Pr}_j[\cdot]$ denote the expectation and probability operators under the instance with $a_J = a_j$. Let $\mathbb{E}_{\text{unif}}[\cdot]$ and $\text{Pr}_{\text{unif}}[\cdot]$ denote the expectation and probability operators under the uniform instance.

Fix an algorithm $\mathcal{A}$. Let $\pi^k$ denote the policy taken by algorithm $\mathcal{A}$ in episode $k$. Let $N_{s_n,a_j} = \sum_{k=1}^{K} \mathbb{1}\left\{\pi^k(s_n) = a_j\right\}$ denote the number of episodes that the policy chooses $a_j$ in state $s_n$. Let $V_{s_n,a_j}$ denote the number of episodes that the algorithm $\mathcal{A}$ visits $(s_n, a_j)$. Let $w(s_n)$ denote the probability of visiting $s_n$ in an episode (the probability of visiting $s_n$ is the same for all policies). Then, it holds that $\mathbb{E}[V_{s_n}, a_j] = w(s_n) \cdot \mathbb{E}[N_{s_n,a_j}]$.

According to the definition of nested-CVaR-PbRL risk objective in Eq. 4, we have:

$$V_1^*(s_1) = \frac{(\alpha - \eta) \cdot 0.2 + \eta \cdot 0.8}{\alpha} B\rho, \quad (104)$$

and summing over all episodes $k \in [K]$, we have

$$\mathbb{E}_j[\mathcal{R}(K)] = \sum_{k=1}^{K} \left(2V_1^*(s_1) - V_1^{\pi_{1,k}}(s_1) - V_1^{\pi_{2,k}}(s_1)\right) \quad (105)$$

$$= \frac{1}{A} \sum_{j=1}^{A} \frac{\eta B\rho}{\alpha} \cdot 0.6 \left(2K - \mathbb{E}_j[N_{s_n,a_j}]\right) \quad (106)$$

$$= 0.6 \cdot \frac{\eta B\rho}{\alpha} \cdot \left(K - \frac{1}{A} \sum_{j=1}^{A} \mathbb{E}_j[N_{s_n,a_j}]\right) \quad (107)$$

Therefore, we have

$$\mathbb{E}[\mathcal{R}(K)] = \frac{1}{A} \sum_{j=1}^{A} \sum_{k=1}^{K} \left( V_1^*(s_1) - V_1^{\pi^k}(s_1) \right) \tag{108}$$

$$= \frac{1}{A} \sum_{j=1}^{A} \frac{\eta}{\alpha} \cdot 0.6 B \rho \left( K - \mathbb{E}_j \left[ N_{s_n, a_j} \right] \right) \tag{109}$$

$$= 0.6 B \rho \cdot \frac{\eta}{\alpha} \cdot \left( K - \frac{1}{A} \sum_{j=1}^{A} \mathbb{E}_j \left[ N_{s_n, a_j} \right] \right) \tag{110}$$

For any $j \in \{A, B\}$, using Pinsker's inequality and $0 < \alpha < \frac{1}{3}$, we have that $\mathrm{KL}\left( p_{\mathrm{unif}}(s_n, \pi_j(s_n)) \| p_j(s_n, \pi_j(s_n)) \right) = \mathrm{KL}(\mathrm{Ber}(\alpha) \| \mathrm{Ber}(\alpha - \eta)) \leq \frac{\eta^2}{(\alpha - \eta)(1 - \alpha + \eta)} \leq \frac{c_1 \eta^2}{\alpha}$ for some constant $c_1$ and small enough $\eta$. Then, using Lemma A. 1 in Du et al. [2022], we have ,

$$\mathbb{E}_j \left[ N_{\pi_j} \right] \leq \mathbb{E}_{\mathrm{unif}} \left[ N_{\pi_j} \right] + \frac{K}{2} \sqrt{\mathbb{E}_{\mathrm{unif}} \left[ V_{\pi_j} \right] \cdot \mathrm{KL}\left( \pi_j(s_n)) \| p_j(s_n, \pi_j(s_n) \right)} \tag{111}$$

$$\leq \mathbb{E}_{\mathrm{unif}} \left[ N_{\pi_j} \right] + \frac{K}{2} \sqrt{w(s_n) \cdot \mathbb{E}_{\mathrm{unif}} \left[ N_{\pi_j} \right] \cdot \frac{c_1 \eta^2}{\alpha}} \tag{112}$$

Then, Using $\sum_{j=1}^{A} \mathbb{E}_{unif} \left[ N_{s_n, a_j} \right] = 2K$ and Cauchy–Schwarz inequality, we have:

$$\frac{1}{A} \sum_{j=1}^{A} \mathbb{E}_j \left[ N_{s_n, a_j} \right] \leq \frac{1}{A} \sum_{j=1}^{A} \mathbb{E}_{unif} \left[ N_{s_n, a_j} \right] + \frac{K\eta}{2A} \sum_{j=1}^{A} \sqrt{\frac{c_1}{\alpha} \cdot w(s_n) \cdot \mathbb{E}_{unif} \left[ N_{s_n, a_j} \right]} \tag{113}$$

$$\leq \frac{1}{A} \sum_{j=1}^{A} \mathbb{E}_{unif} \left[ N_{s_n, a_j} \right] + \frac{K\eta}{2A} \sqrt{A \sum_{j=1}^{A} \frac{c_1}{\alpha} \cdot w(s_n) \cdot \mathbb{E}_{unif} \left[ N_{s_n, a_j} \right]} \tag{114}$$

$$\leq \frac{K}{A} + \frac{K\eta}{2} \sqrt{\frac{c_1 \cdot w(s_n) K}{\alpha A}} \tag{115}$$

Thus, we have :

$$\mathbb{E}[\mathcal{R}(K)] \geq 0.6 B \rho \cdot \frac{\eta}{\alpha} \cdot \left( K - \frac{K}{A} - \frac{K\eta}{2} \sqrt{\frac{c_1 \cdot w(s_n) K}{\alpha A}} \right). \tag{116}$$

Let $\eta = c_2 \sqrt{\frac{\alpha A}{w(s_n) K}}$ for a small enough constant $c_2$. We have

$$\mathbb{E}[\mathcal{R}(K)] = \Omega \left( B \rho \sqrt{\frac{A}{\alpha \cdot w(s_n) K}} \cdot K \right)$$

$$= \Omega \left( B \rho \sqrt{\frac{AK}{\alpha \cdot w(s_n)}} \right)$$

**Hard to learn case 2.**

The trajectory reward functions are as follows. For any $\xi_H \in \mathcal{Z}_H$, $\phi(\xi_H) = (\mathbb{I}(S_H(\xi_H) = x_1)B, \mathbb{I}(S_H(\xi_H) = x_2)B, \mathbb{I}(S_H(\xi_H) = x_3)B, \mathbb{I}(S_H(\xi_H) = x_4)B)$, $w_r = (1\rho, 0.8\rho, 0.2\rho, (0.2 - \eta)\rho)$

The transition distributions are as follows. Let $\mu$ be a parameter which satisfies that $0 < \alpha < \mu < \frac{1}{3}$. For any $a \in \mathcal{A}, p(s_2 \mid s_1, a) = \mu, p(x_1 \mid s_1, a) = 1 - 3\mu, p(x_2 \mid s_1, a) = \mu$ and $p(x_3 \mid s_1, a) = \mu$. For any $i \in \{2, \ldots, n-1\}$ and $a \in \mathcal{A}, p(s_{i+1} \mid s_i, a) = \mu$ and $p(x_1 \mid s_i, a) = 1 - \mu.x_1, x_2$ and $x_3$ are absorbing states, i.e., for any $a \in \mathcal{A}, p(x_1 \mid x_1, a) = 1, p(x_2 \mid x_2, a) = 1$ and $p(x_3 \mid x_3, a) = 1$.

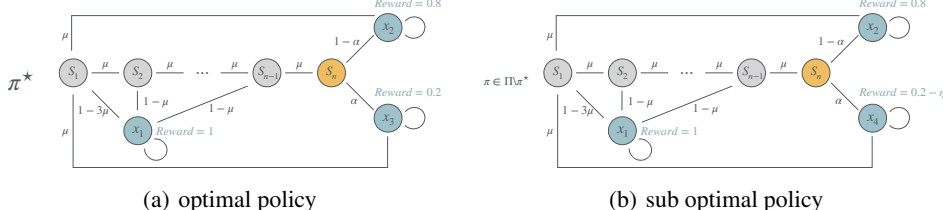

(a) optimal policy    (b) sub optimal policy

Figure 6: Hard to learn case 2

$p\left(x_2 \mid s_n, a_J\right) = 1 - \alpha$ and $p\left(x_3 \mid s_n, a_J\right) = \alpha$, where $\eta$ is a parameter which satisfies $0 < \eta < \alpha$ and will be chosen later. For any suboptimal action $a \in \mathcal{A} \setminus \{a_J\}$, $p\left(x_2 \mid s_n, a\right) = 1 - \alpha$ and $p\left(x_4 \mid s_n, a\right) = \alpha$ For any $a_j \in \mathcal{A}$, let $\mathbb{E}_j[\cdot]$ and $\Pr_j[\cdot]$ denote the expectation and probability operators under the instance with $a_J = a_j$. Let $\mathbb{E}_{\text{unif}}[\cdot]$ and $\Pr_{\text{unif}}[\cdot]$ denote the expectation and probability operators under the uniform instance.

According to the definition of nested-CVaR-PbRL risk objective in 4, we have:

$$V_1^*\left(s_1\right) = 0.2\eta B \rho, \tag{117}$$

Thus,

$$\mathbb{E}_j[\mathcal{R}(K)] = \sum_{k=1}^{K} \left(2V_1^*\left(s_1\right) - V_1^{\pi_{1,k}}\left(s_1\right) - V_1^{\pi_{2,k}}\left(s_1\right)\right) \tag{118}$$

$$= \frac{1}{A} \sum_{j=1}^{A} 0.2\eta B \rho \left(2K - \mathbb{E}_j\left[N_{s_n, a_j}\right]\right) \tag{119}$$

$$= 0.2\eta B \rho \cdot \left(K - \frac{1}{A} \sum_{j=1}^{A} \mathbb{E}_j\left[N_{s_n, a_j}\right]\right) \tag{120}$$

Therefore, we have

$$\mathbb{E}[\mathcal{R}(K)] = \frac{1}{A} \sum_{j=1}^{A} \sum_{k=1}^{K} \left(V_1^*\left(s_1\right) - V_1^{\pi^k}\left(s_1\right)\right) \tag{121}$$

$$= \frac{1}{A} \sum_{j=1}^{A} 0.2\eta B \rho \rho \left(K - \mathbb{E}_j\left[N_{s_n, a_j}\right]\right) \tag{122}$$

$$= 0.2\eta B \rho \cdot \frac{\eta}{\alpha} \cdot \left(K - \frac{1}{A} \sum_{j=1}^{A} \mathbb{E}_j\left[N_{s_n, a_j}\right]\right) \tag{123}$$

For any $j \in \{A, B\}$, since the preference based on Bernoulli distribution, using Pinsker's inequality and $0 < \alpha < \frac{1}{3}$, we have that $\text{KL}\left(p_{\text{unif}}\left(s_n, \pi_j(s_n)\right) \| p_j\left(s_n, \pi_j(s_n)\right)\right) = \text{KL}(\text{Ber}(\alpha) \| \text{Ber}(\alpha - \eta)) \leq \frac{\eta^2}{(0.2-\eta)(0.8+\eta)} \leq \frac{c_1 \eta^2}{0.16}$ for some constant $c_1$ and small enough $\eta$. Then, we have ,

$$\mathbb{E}_j\left[N_{\pi_j}\right] \leq \mathbb{E}_{\text{unif}}\left[N_{\pi_j}\right] + \frac{K}{2} \sqrt{\mathbb{E}_{\text{unif}}\left[V_{\pi_j}\right] \cdot \text{KL}\left(\pi_j(s_n) \| p_j\left(s_n, \pi_j(s_n)\right)\right)} \tag{124}$$

$$\leq \mathbb{E}_{\text{unif}}\left[N_{\pi_j}\right] + \frac{K}{2} \sqrt{w\left(s_n\right) \cdot \mathbb{E}_{\text{unif}}\left[N_{\pi_j}\right] \cdot \frac{c_1 \eta^2}{0.16}} \tag{125}$$

Then, Using $\sum_{j=1}^{A} \mathbb{E}_{unif}\left[N_{s_n, a_j}\right] = 2K$ and Cauchy–Schwarz inequality, we have:

$$\frac{1}{A}\sum_{j=1}^{A}\mathbb{E}_j\left[N_{s_n,a_j}\right] \leq \frac{1}{A}\sum_{j=1}^{A}\mathbb{E}_{unif}\left[N_{s_n,a_j}\right] + \frac{K\eta}{2A}\sum_{j=1}^{A}\sqrt{\frac{c_1}{0.16}\cdot w\left(d\right)\cdot\mathbb{E}_{unif}\left[N_{s_n,a_j}\right]} \quad (126)$$

$$\leq \frac{1}{A}\sum_{j=1}^{A}\mathbb{E}_{unif}\left[N_{s_n,a_j}\right] + \frac{K\eta}{2A}\sqrt{A\sum_{j=1}^{A}\frac{c_1}{0.16}\cdot w\left(d\right)\cdot\mathbb{E}_{unif}\left[N_{s_n,a_j}\right]} \quad (127)$$

$$\leq \frac{K}{A} + \frac{K\eta}{2}\sqrt{\frac{c_1\cdot w\left(d\right)K}{0.16A}} \quad (128)$$

Thus, we have :

$$\mathbb{E}[\mathcal{R}(K)] \geq 0.2B\rho\cdot\eta\cdot\left(K - \frac{K}{A} - \frac{K\eta}{2}\sqrt{\frac{c_1\cdot w\left(d\right)K}{0.16A}}\right). \quad (129)$$

Let $\eta = c_2\sqrt{\frac{0.16A}{w(d)K}}$ for a small enough constant $c_2$. We have

$$\mathbb{E}[\mathcal{R}(K)] = \Omega\left(B\rho\sqrt{\frac{A}{w\left(d\right)K}}\cdot K\right)$$

$$= \Omega\left(B\rho\sqrt{\frac{AK}{w\left(d\right)}}\right)$$

$\square$

# F  The optimal policy calculation for known PbRL MDP

In this section, we describe how to compute the optimal policy for the CVaR objective when the PbRL-MDP is known; this approach is described in detail in Bäuerle and Ott [2011], Bastani et al. [2022]. Following this work, we consider the setting where we are trying to minimize cost rather than maximize reward. In particular, consider an know PbRL MDP $\mathcal{M}(\mathcal{S}, \mathcal{A}, r_\xi^\star, \mathbf{P}^\star, H)$, where $\mathcal{S}$ and $\mathcal{A}$, and our goal is to compute a policy $\pi$ that maximizes its CVaR objective.

**Lemma F.1.** *CVaR definition.* *For any random variable Z, we have*

$$\mathrm{CVaR}_\alpha(Z) = \sup_{\rho\in\mathbb{R}}\left\{\rho - \frac{1}{\alpha}\cdot\mathbb{E}_Z\left[(\rho - Z)^+\right]\right\},$$

*where the minimum is achieved by* $\rho^* = \mathrm{VaR}_\alpha(Z)$.

## F.1  Static CVaR object

Since the optimal policy for static CVaR object satisfies:

$$\pi^\star \in \mathrm{argmax}_{\pi\in\Pi}\,\mathrm{CVaR}(Z(\pi)) \quad (130)$$

As a consequence of Lemma F.1, we have

$$\mathrm{CVaR}\left(Z^{(\pi^\star)}\right) = \max_{\pi\in\Pi}\mathrm{CVaR}\left(Z^{(\pi)}\right) \quad (131)$$

$$= \max_{\pi\in\Pi}\sup_{\rho\in\mathbb{R}}\left\{\rho - \frac{1}{\alpha}\cdot\mathbb{E}_{Z}(\pi)\left[\left(\rho - Z^{(\pi)}\right)^+\right]\right\} \quad (132)$$

$$= \sup_{\rho\in\mathbb{R}}\left\{\rho - \frac{1}{\alpha}\cdot\max_{\pi\in\Pi}\mathbb{E}_{Z^{(\pi)}}\left[\left(\rho - Z^{(\pi)}\right)^+\right]\right\}. \quad (133)$$

Thus, the optimal policy is:

$$\pi^\star = \arg\max_{\pi\in\Pi}\mathbb{E}_{Z^{(\pi)}}\left[\left(\rho^\star - Z^{(\pi)}\right)^+\right] \tag{134}$$

where

$$\rho^* = \arg\sup_{\rho\in\mathbb{R}}J(\rho) \quad \text{where} \quad J(\rho) = \rho - \frac{1}{\alpha}\cdot\max_{\pi\in\Pi}\mathbb{E}_Z(\pi)\left[\left(\rho - Z^{(\pi)}\right)^+\right]. \tag{135}$$

**Value iteration.** we reconstruct the MDP as enlarge the state space $\tilde{s}_h = (\xi_h, \rho)$, where $\rho$ will work as a quantile value. Letting $S_1$ be the initial state of the original PbRL MDP $\mathcal{M}$ .

We iterate the value and calculate the policy as follows:

$$\widetilde{V}_H((\xi_H, \rho)) = \max\{\rho - r_\xi^\star(\xi_H), 0\} \tag{136}$$

$$\widetilde{V}_h((\xi_h, \rho)) = \max_{a\in A}\int \widetilde{V}_{h+1}\left((s'\circ(\xi_h, a), \rho)\right)\cdot\mathbf{P}^\star\left(s' \mid (S_h(\xi_h), a)\right) \tag{137}$$

$$\pi(\xi_h) = argmax_{a\in A}\int \widetilde{V}_{h+1}\left((s'\circ(\xi_h, a), \rho)\right)\cdot\mathbf{P}^\star\left(s' \mid (S_h(\xi_h), a)\right) \tag{138}$$

Then, given an initial state $s_1$, we construct state $\Im_1 = (s_1, -\rho^*)$, where

$$\rho^* = argsup_{\rho\in\mathbb{R}}\left\{\rho - \frac{1}{\alpha}\cdot\widetilde{V}_1^{(\pi)}((s_1, -\rho))\right\},$$

and then acting optimally in $\mathcal{M}$.

### F.2 Nested CVaR object

According to Eq. 4, nested CVaR object could directly use the Bellman equation to iterate the value.

**Value iteration.** we reconstruct the MDP as enlarge the state space $\tilde{s}_h = (\xi_h, \rho)$, where $\rho$ will work as a quantile value. Letting $S_1$ be the initial state of the original PbRL MDP $\mathcal{M}$ .

We iterate the value and calculate the policy as follows:

$$\widetilde{V}_H((\xi_H, \rho)) = r_\xi^\star(\xi_H) \tag{139}$$

$$\widetilde{V}_h((\xi_h, \rho)) = \max_{\pi\in\Pi}\sup_{\rho\in\mathbb{R}}\left\{\rho - \frac{1}{\alpha}\cdot\mathbb{E}_{\widetilde{V}_{h+1}((s'\circ(\xi_h, a), \rho))}\left[\left(\rho - \widetilde{V}_{h+1}\left((s'\circ(\xi_h, a), \rho)\right)\right)^+\right]\right\} \tag{140}$$

$$\pi(\xi_h) = \arg\max_{\pi\in\Pi}\sup_{\rho\in\mathbb{R}}\left\{\rho - \frac{1}{\alpha}\cdot\mathbb{E}_{\widetilde{V}_{h+1}((s'\circ(\xi_h, a), \rho))}\left[\left(\rho - \widetilde{V}_{h+1}\left((s'\circ(\xi_h, a), \rho)\right)\right)^+\right]\right\} \tag{141}$$

then acting optimally in $\mathcal{M}$.

## G   AUXILIARY LEMMAS

**Definition G.1.** $\alpha$**-dependence in Russo and Van Roy [2013]**. For $\alpha > 0$ and function class $\mathcal{Z}$ whose elements are with domain $\mathcal{X}$, an element $x \in \mathcal{X}$ is $\alpha$-dependent on the set $\mathcal{X}_n := \{x_1, x_2, \cdots, x_n\} \subset \mathcal{X}$ with respect to $\mathcal{Z}$, if any pair of functions $z, z' \in \mathcal{Z}$ with $\sqrt{\sum_{i=1}^n (z(x_i) - z'(x_i))^2} \leqslant \alpha$ satisfies $z(x) - z'(x) \leqslant \alpha$. Otherwise, $x$ is $\alpha$-independent on $\mathcal{X}_n$ if it does not satisfy the condition.

**Definition G.2. Eluder dimension in Russo and Van Roy [2013]**. For $\alpha > 0$ and function class $\mathcal{Z}$ whose elements are with domain $\mathcal{X}$, the Eluder dimension $\dim_E(\mathcal{Z}, \alpha)$, is defined as the length of the longest possible sequence of elements in $\mathcal{X}$ such that for some $\alpha' \geqslant \alpha$, every element is $\alpha'$ independent of its predecessors.

**Definition G.3. Covering number** Given two functions $l$ and $u$, the bracket $[l, u]$ is the set of all functions $f$ satisfying $l \leq f \leq u$. An $\alpha$-bracket is a bracket $[l, u]$ with $\|u - l\| < \alpha$. The covering number $N_{[\cdot]}(\mathcal{F}, \alpha, \|\cdot\|)$ is the minimum number of $\alpha$-brackets needed to cover $\mathcal{F}$.

**Lemma G.4.** *(Linear Preference Models Eluder dimension and Covering number). For the case of $d$-dimensional generalized trajectory linear feature models $r_\xi(\xi_H) := \langle \phi(\xi_H), \mathbf{w}_r \rangle$, where $\phi : Traj \to \mathbb{R}^{dim_{\mathbb{T}}}$ is a known $dim_{\mathbb{T}}$ dimension feature map satisfying $\|\psi(\xi_H)\|_2 \leq B$ and $\theta \in \mathbb{R}^d$ is an unknown parameter with $\|\mathbf{w}_r\|_2 \leq \rho_w$. Then the $\alpha$-Eluder dimension of $r_\xi(\xi_H)$ is at most $\mathcal{O}(dim_{\mathbb{T}} \log(B\rho_w/\alpha))$. The $\alpha$ - covering number is upper bounded by $\left(\frac{1+2B\rho_w}{\alpha}\right)^{dim_{\mathbb{T}}}$.*

Let $(X_p, Y_p)_{p=1,2,\dots}$ be a sequence of random elements, $X_p \in X$ for some measurable set $X$ and $Y_p \in \mathbb{R}$. Let $\mathcal{F}$ be a subset of the set of real-valued measurable functions with domain $X$. Let $\mathbb{F} = (\mathbb{F}_p)_{p=0,1,\dots}$ be a filtration such that for all $p \geq 1, (X_1, Y_1, \cdots, X_{p-1}, Y_{p-1}, X_p)$ is $\mathbb{F}_{p-1}$ measurable and such that there exists some function $f_\star \in \mathcal{F}$ such that $\mathbb{E}[Y_p \mid \mathbb{F}_{p-1}] = f_*(X_p)$ holds for all $p \geq 1$. The (nonlinear) least square predictor given $(X_1, Y_1, \cdots, X_t, Y_t)$ is $\hat{f}_t = \arg\min_{f \in \mathcal{F}} \sum_{p=1}^t (f(X_p) - Y_p)^2$. We say that $Z$ is conditionally $\rho$-subgaussian given the $\sigma$-algebra $\mathbb{F}$ is for all $\lambda \in \mathbb{R}, \log \mathbb{E}[\exp(\lambda Z) \mid \mathbb{F}] \leq \frac{1}{2}\lambda^2 \rho^2$. For $\alpha > 0$, let $N_\alpha$ be the $\|\cdot\|_\infty$-covering number of $\mathcal{F}$ at scale $\alpha$. For $\beta > 0$, define

$$\mathcal{F}_t(\beta) = \left\{ f \in \mathcal{F} : \sum_{p=1}^t \left( f(X_p) - \hat{f}_t(X_p) \right)^2 \leq \beta \right\}. \tag{142}$$

**Lemma G.5.** *(Theorem 5 of Ayoub et al. [2020]). Let $\mathbb{F}$ be the filtration defined above and assume that the functions in $\mathcal{F}$ are bounded by the positive constant $C > 0$. Assume that for each $s \geq 1, (Y_p - f_*(X_p))$ is conditionally $\sigma$-subgaussian given $\mathbb{F}_{p-1}$. Then, for any $\alpha > 0$, with probability $1 - \delta$, for all $t \geq 1, f_* \in \mathcal{F}_t(\beta_t(\delta, \alpha))$, where*

$$\beta_t(\delta, \alpha) = 8\sigma^2 \log(2N_\alpha/\delta) + 4t\alpha \left( C + \sqrt{\sigma^2 \log(4t(t+1)/\delta)} \right).$$

**Lemma G.6.** *(Lemma 5 of Russo and Van Roy [2013]). Let $\mathcal{F} \in B_\infty(X, C)$ be a set of functions bounded by $C > 0$, $(\mathcal{F}_t)_{t \geq 1}$ and $(x_t)_{t \geq 1}$ be sequences such that $\mathcal{F}_t \subset \mathcal{F}$ and $x_t \in \mathcal{X}$ hold for $t \geq 1$. Let $\mathcal{F}|_{x_{1:t}} = \{(f(x_1), \dots, f(x_t)) : f \in \mathcal{F}\} (\subset \mathbb{R}^t)$ and for $S \subset \mathbb{R}^t$, let $\text{diam}(S) = \sup_{u,v \in S} \|u - v\|_2$ be the diameter of $S$. Then, for any $T \geq 1$ and $\alpha > 0$ it, holds that*

$$\sum_{t=1}^T \text{diam}\left(\mathcal{F}_t|_{x_t}\right) \leq \alpha + C(d \wedge T) + 2\delta_T \sqrt{dT},$$

*where $\delta_T = \max_{1 \leq t \leq T} \text{diam}\left(\mathcal{F}_t|_{x_{1:t}}\right)$ and $d = \dim_{\mathcal{E}}(\mathcal{F}, \alpha)$.*

**Lemma G.7.** *If $(\beta_t \geq 0 \mid t \in \mathbb{N})$ is a nondecreasing sequence and $\mathcal{F}_t := \left\{ f \in \mathcal{F} : \left\| f - \hat{f}_t^{LS} \right\|_{2,E_t} \leq \sqrt{\beta_t} \right\}$, where $\hat{f}_t^{LS} \in \arg\min_{f \in \mathcal{F}} L_{2,t}(f)$ and $L_{2,t}(f) = \sum_1^{t-1} (f(A_t) - R_t)^2$, then for all $T \in \mathbb{N}$ and $\epsilon > 0$,*

$$\sum_{t=1}^T \mathbf{1}\left(w_{\mathcal{F}_t}(A_t) > \epsilon\right) \leq \left( \frac{4\beta_T}{\epsilon^2} + 1 \right) \dim_E(\mathcal{F}, \epsilon)$$

*where $w_{\mathcal{F}}(a) := \sup_{f \in \mathcal{F}} f(a) - \inf_{f \in \mathcal{F}} f(a)$ denotes confidence interval widths.*

**Theorem G.8.** *Hoeffding's inequality[Hoeffding, 1994]. Let $X_1, X_2, \dots, X_n$ be independent random variables that are sub-Gaussian with parameter $\sigma$. Define $S_n = \sum_{i=1}^n X_i$. Then, for any $t > 0$, Hoeffding's inequality provides an upper bound on the tail probabilities of $S_n$, which is given by:*

$$\Pr(|S_n - \mathbb{E}[S_n]| \geq t) \leq 2 \exp\left( -\frac{t^2}{2n\sigma^2} \right).$$

*This result emphasizes the robustness of the sum $S_n$ against deviations from its expected value, particularly useful in applications requiring high confidence in estimations from independent sub-Gaussian observations.*

**Lemma G.9.** *(Lemma F.4. in Dann et al. [2017]) Let $\mathcal{F}_i$ for $i = 1\ldots$ be a filtration and $X_1, \ldots X_n$ be a sequence of Bernoulli random variables with $\mathbb{P}\left(X_i = 1 \mid \mathcal{F}_{i-1}\right) = P_i$ with $P_i$ being $\mathcal{F}_{i-1}$-measurable and $X_i$ being $\mathcal{F}_i$ measurable. It holds that*

$$\mathbb{P}\left(\exists n : \sum_{t=1}^{n} X_t < \sum_{t=1}^{n} P_t/2 - W\right) \leq e^{-W}$$

