# OpenReview forum: "RA-PbRL: Provably Efficient Risk-Aware Preference-Based Reinforcement Learning"
_NeurIPS.cc/2024/Conference — NeurIPS 2024 poster_

### Official Review · Reviewer_LP4v · 2024-07-08

**Soundness:** 3
**Presentation:** 2
**Contribution:** 2
**Rating:** 4
**Confidence:** 3

**Summary:**

This paper studies and proves the applicability of two risk-aware objectives to Preference-Based Reinforcement Learning (PbRL), i.e., iterated and accumulated quantile risk objectives. The authors design an algorithm called Risk-Aware-PbRL (RA-PbRL), which can optimize both iterated and accumulated objectives. Furthermore, the authors provide a theoretical analysis of the regret bounds. The results demonstrate that the regret bounds of algorithm RA-PbRL under both the iterated and accumulated objectives are sublinear with respect to the number of episodes. Finally, the authors present empirical results to support their theoretical findings.

**Strengths:**

1.	The studied problem, i.e., applying the iterated and accumulated risk-aware objectives to PbRL, is relevant and useful for some applications such as healthcare and AI systems. The considered reward model depends on the feature of the trajectory, instead of the feature of the state-action pair, which is more general than the prior works.
2.	The authors consider both the iterated and accumulated risk-aware objectives, which encompass the popular CVaR objective. In addition, the authors also design an algorithm and provide regret analysis for both objectives. The algorithm design and theoretical analysis are well executed.

**Weaknesses:**

1.	The proposed algorithm is very straightforward, which seems to simply combine the confidence set construction and the risk-aware objective. This algorithm is computationally inefficient and hard to implement in practice. Can the authors explain more on how to implement this algorithm?
2.	What is the intuition behind the factor $\min_{\pi,d} \omega_{\pi}(d)$ in Theorem 4.1? In particular, why the probability that the feature is not zero will influence the regret? More discussions on the regret due to reward estimation are needed.
3.	It seems that $L_G$ appears in the upper bound (Theorem 4.2) but not in the lower bound (Theorem 4.4). Why the authors said that “it demonstrates the significant impact of LG” below Theorem 4.4? In addition, it seems that $dim_{T}$ appears in the lower bound, but not in the upper bound. Can the authors explain more on it?
4.	This paper needs careful proof-reading. There are many typos. For example, the factor $\min_{\pi,d} \omega_{\pi}(d)$ in Theorems 4.1 and 4.3, and the $O(…)$ notation in Theorems 4.1 and 4.2. In Line 130, “use” should be “used”. In Line 197, the “and” should be moved to the front of “$V^{\pi}_i$ in Eq.5 …”?

**Questions:**

Please see the weaknesses above.

**Limitations:**

Please see the weaknesses above.

---

> ### Author Rebuttal · Authors · 2024-08-07
>
> We sincerely thank the reviewer for the effort in reviewing our paper and appreciate our versatility compared to prior work. The following are responses to the reviewer’s concern.
>
> > Weakness:  The proposed algorithm is very straightforward, which seems to simply combine the confidence set construction and the risk-aware objective. This algorithm is computationally inefficient and hard to implement in practice. Can the authors explain more about how to implement this algorithm?
>
> Thank you for your feedback.We have established a formal algorithmic framework with well-defined regret bounds for the PbRL problem. This "straightforward" framework draws inspiration from numerous RL theory papers, such as PbRL by Pacchiano et al., 2023, and risk-aware RL by Bastani et al., 2022. Our main technical challenges lie in estimating the confidence set and proving the regret bounds. We have employed rigorous mathematical techniques for these estimations, including methods like least squares and the application of covering numbers. In practice, to address your concerns about computational efficiency and implementation, we have successfully implemented this algorithm in a Mujoco environment demonstrating good performance. Further implementation details, particularly concerning the construction of confidence sets and integration with risk-aware objectives, are provided in public comments aid practical application.
>
>
> > Weakness2: What is the intuition behind the factor $\min _{\pi, d} \omega_\pi(d)$ in Theorem 4.1? In particular, why the probability that the feature is not zero will influence the regret? More discussions on the regret due to reward estimation are needed.
>
> Thank you for raising this insightful question. We acknowledge the need for more discussion on how reward estimation affects regret and will enhance our paper with a detailed explanation.  The factor $\min *{\pi, d} \omega*\pi(d)$ in Theorem 4.1 plays a critical role, which influences regret through the exploration of critical but seldom visited states. In particular, Appendix E Figure 4 illustrates a case where states with less possibility to vist, due to their higher uncertainty in estimation, lead to higher regret. This addition will clarify the intuition behind information entropy and its impact on regret calculations.
>
>
> > Weakness 3: It seems that L_G appears in the upper bound (Theorem 4.2) but not in the lower bound (Theorem 4.4). Why did the authors say that "it demonstrates the significant impact of LG" below Theorem 4.4? In addition, it seems that \operatorname{dim}_T appears in the lower bound, but not in the upper bound. Can the authors explain more on it?
>
> Thank you for your concern. Indeed, a more detailed explanation of the regret results is warranted. The reason we mention "it demonstrates the significant impact of L_G" is as follows: By comparing the lower bound to the upper bound, the presence of $L_G $highlights the unique challenges in risk-aware settings. Reducing the gap between these bounds is challenging because $L_G$, derived from the quantile function, assigns weights to the $\alpha$-tile states, thereby amplifying the effects of estimation errors.
>
> Regarding the issue with $\operatorname{dim}_T$, there was an oversight in our documentation. Theorem 4.2 actually pertains specifically to tabular settings. The more general result is found on Page 26, Lemma D.21. We will address this discrepancy and make the necessary corrections in the revised manuscript.
>
> > Weakness 4: This paper needs careful proofreading. There are many typos. For example, the factor $\min_{\pi, d} \omega_\pi(d)$ in Theorems 4.1 and 4.3, and the $O(\ldots)$ notation in Theorems 4.1 and 4.2. In Line 130, "use" should be "used". In Line 197, the "and" should be moved to the front of " $V_i^\pi$ in Eq. 5 ..."?
>
> We appreciate your meticulous review and will carefully proofread the manuscript to correct these and any other typos.

---

> > ### Comment · Reviewer_LP4v · 2024-08-11
> > **Thank the authors for their response**
> >
> > Thank the authors for their response. I tend to maintain my score.

---

> > > ### Author Response · Authors · 2024-08-13
> > >
> > > Thank you for acknowledging our response. We would like to check if our rebuttal addressed the raised concerns. We are happy to address any specific questions remaining.

---

### Official Review · Reviewer_EoFm · 2024-07-12

**Soundness:** 3
**Presentation:** 2
**Contribution:** 2
**Rating:** 7
**Confidence:** 2

**Summary:**

This paper incorporates risk-awareness into Preference-based Reinforcement Learning (PbRL). Specifically, it tackles the issue that under PbRL, the reward is episodic, meaning that it can only be computed on full trajectories. The authors adapt both iterative and accumulated quantile risk objectives to deal with episodic rewards.

Additionally, the paper presents an algorithm (RA-PbRL) to incorporate these objectives into PbRL.

Lastly, the authors provide regret boundaries for both iterative and accumulated quantile risk objectives with RA-PbRL.

**Strengths:**

[Quality, Clarity] The paper is well written (particularly the introduction). As far as I could follow the mathematical development is robust, with regret upper- and lower-bounds being established for both types of risk considered.

**Weaknesses:**

*  **W1** [Quality]: As the manuscript already mentions, the experimental setting is very simple. It would have been interesting to train more complex settings, particular from actual human preferences.
 *  **W2** [Significance]: I think more could be done to stress the important of risk-awareness in the PbRL setting. After reading the paper, it was still not clear to me which applications would benefit from RA-PbRL.

**Questions:**

In PbRL, it is often assumed that the given preferences $o_i$ may be noisy. Do authors assume perfect preferences? If not, it would be interesting to analyse how the amount of noise in the preference affects the regret bounds.

**Limitations:**

The limitations have been correctly addressed.

---

> ### Author Rebuttal · Authors · 2024-08-07
>
> We are grateful to the reviewer for the positive score and for recognizing the mathematical development of our work. We summarize our responses to the concerns and revisions that we will make to this paper.
>
> > Weakness 1 : As the manuscript already mentions, the experimental setting is very simple. It would have been interesting to train more complex settings, particularly from actual human preferences.
>
> We have added experiments conducted in Mujoco, a commonly used simulation environment in robotics and deep reinforcement learning (see public comments and attachment), where our theory-based model performed well, demonstrating the potential of our work to provide useful guidance in real-world problems. We also agree that real human preferences constitute a good experimental setting, and we plan to implement our algorithm in this setting in the future.
>
>
> > Weakness2 [Significance]: I think more could be done to stress the important of risk-awareness in the PbRL setting. After reading the paper, it was still not clear to me which applications would benefit from RA-PbRL.
>
> A2:  Thank you for your insightful suggestion. RA-PbRL (Risk-Aware Preference-based Reinforcement Learning) has significant potential for various real-life applications. For instance, in autonomous driving, PbRL can reduce the computational burden by obviating the need to calculate reward signals from every state-action pair, while incorporating risk-awareness to enhance safety (referenced in Chen et al., 2022). Additionally, in fields like generative AI, including systems like ChatGPT (discussed in Zhuo et al., 2023; Qi et al., 2023; Chen et al., 2023), PbRL is utilized to gauge algorithm performance where managing risk is crucial to prevent the generation of harmful content. We agree that emphasizing these applications could better highlight the importance of our contributions, and we will expand on this aspect in our revised manuscript.
> > Question1: In PbRL , it is often assumed that the given preferences o_i may be noisy. Do authors assume perfect preferences? If not, it would be interesting to analyse how the amount of noise in the preference affects the regret bounds.
>
> In our framework, we do not assume perfect preferences; instead, we model the given preferences o_i as adhering to a Bernoulli distribution, as stated on line 137 of our paper. This setup involves a noisy comparison (duel) between two trajectories, consistent with the settings used in Pacchiano et al., 2023, and Chen et al., 2022. Assuming noise in the utility or reward leads to similar outcomes for preferences since the Bernoulli distribution remains invariant.

---

> > ### Comment · Reviewer_EoFm · 2024-08-12
> >
> > I thank the authors for their detailed rebuttal and extended experiments. Since authors adequately addressed all my questions and concerns I will raise my score to a 7.

---

> > > ### Author Response · Authors · 2024-08-13
> > >
> > > Thank you for acknowledging our response and updating the score. We appreciate your time and efforts.

---

### Official Review · Reviewer_QEr2 · 2024-07-13

**Soundness:** 3
**Presentation:** 3
**Contribution:** 3
**Rating:** 5
**Confidence:** 2

**Summary:**

This paper focuses on the theoretical analysis of risk-aware preference-based reinforcement learning and introduces Risk-Aware-PbRL (RA-PbRL) to optimize both iterated and accumulated risk-aware objectives.

**Strengths:**

- This paper proves that both iterated and accumulated quantile risk-aware objectives can be applied to one-episode reward settings. This may provide a theoretical foundation for future episodic RL or PbRL methods focusing on risk-related objectives.
- This paper provide analysis of regret guarantee on the proposed risk-aware algorithm.

**Weaknesses:**

- I think it is necessary to clearly clarify all the strong assumptions, not only the linear reward function assumption, as these strong assumptions typically cannot be met in real-world control scenarios. This would make the theoretical results in this paper more applicable and useful for researchers using deep PbRL methods to address real-world control problems.
- Why is the regret of RA-PbRL higher than PbOP when $\alpha$ is small (Fig. 1(b) and 2(b))? Can the authors provide an intuitive explanation for this phenomenon?

**Questions:**

See above.

**Limitations:**

None.

---

> ### Author Rebuttal · Authors · 2024-08-07
>
> We appreciate the constructive feedback and valuable time for evaluating our paper. We especially thank the reviewer for mentioning that “provide a theoretical foundation for future episodic RL or PbRL methods focusing on risk-related objectives.” The reviewer has raised some valid questions and provided many mindful suggestions. The following are our responses to the reviewer’s comments:
>
> > Weakness1: I think it is necessary to clearly clarify all the strong assumptions, not only the linear reward function assumption, as these strong assumptions typically cannot be met in real-world control scenarios. This would make the theoretical results in this paper more applicable and useful for researchers using deep PbRL methods to address real-world control problems.
>
> Thank you for your valuable suggestions. In this work, we primarily extend the frameworks set by Pacchiano et al. in their 2022 PbRL work and Bastani et al. in their 2022 risk-aware RL work. The only modification we made is the assumption that the absolute values of components of trajectory embeddings has a gap between 0 and a positive number \( b \) (Assumption 3.1), which is entirely reasonable, especially in the context of a finite-step discrete action space (a common scenario) where its validity is evident. Regarding the practical applications of our work, we have supplemented it with experiments in Mujoco, a commonly used simulation environment in robotics and deep reinforcement learning (see public comments), where our theory-based model performed well, demonstrating the potential of our work to provide useful guidance in real-world problems.
>
> >Weakness2: Why is the regret of RA-PbRL higher than PbOP when alpha is small (Fig. 1(b) and 2(b))? Can the authors provide an intuitive explanation for this phenomenon?
>
> I would like to clarify your question: are you asking why RA-PbRL exhibits higher regret than PbOP in the early episodes under the minimal setting of alpha=0.05, as shown in Fig. 1(a) and 2(a)? Because in the figures you mentioned, Fig. 1(b) and 2(b), our algorithm did not exhibit higher regret.
> Here is an intuitive explanation for why our algorithm exhibits greater regret compared to the non-risk-aware algorithm in the early stages: Unlike PbOP, which solely focuses on maximizing the average reward, our RA-PbRL algorithm emphasizes risk consideration. Imagine a person learning a potentially risky new skill, such as driving or chopping vegetables. They are likely to choose a more conservative strategy with lower expected returns to avoid risks (think about your first driving experience, you couldn’t drive as efficiently as an experienced driver). However, as learning progresses (iterations of the algorithm), and the risk boundaries are gradually mastered, we can maintain high returns while always avoiding substantial losses, thereby surpassing the risk-neutral PbOP after a certain number of episodes. The reason this phenomenon appears only at a small alpha might be because the probability of risk events is low enough that early results can be good even if they are ignored. For larger alphas, risk-neutral algorithms initially fall behind due to these high-risk events.

---

> > ### Author Response · Authors · 2024-08-13
> >
> > We want to express our gratitude once again for your valuable feedback. Since the discussion period is coming to a close, we would be grateful if you could share any additional thoughts you may have about our rebuttal. Thank you for your time.

---

> ### Comment · Reviewer_QEr2 · 2024-08-13
>
> Thank you for your responses.
>
> Yes, it is indeed Fig. 1(a) and 2(a). Your explanation makes sense.
>
> I will maintain my score.

---

> > ### Author Response · Authors · 2024-08-13
> >
> > Thank you for acknowledging our response. We are glad to know our explanation addressed your questions.

---

### Official Review · Reviewer_TVSN · 2024-07-15

**Soundness:** 2
**Presentation:** 3
**Contribution:** 3
**Rating:** 6
**Confidence:** 4

**Summary:**

This paper studies preference-based RL (PbRL) where instead of the expected return, the agent optimizes a risk measure based on preference feedback. The authors study two settings called "iterated" and "accumulated" quantile risks, otherwise known as nested and static risks. They provide sublinear regret bounds for both approaches and instantiate a hard-to-solve MDP to establish a lower bound.

**Strengths:**

The proposed research is well-motivated, as it combines the accessibility of preference-based feedback with risk-sensitive RL, which is crucial for safety-critical applications.

If correct, the contributions are meaningful, as they englobe both nested and static risk measures.

The text has a nice flow. I have to say that I reviewed a previous version of this submission and the structure is much clearer now.

**Weaknesses:**

- Some used terminologies are non-standard for the risk-sensitive RL (RS-RL) community. For example, the authors discuss "iterated" versus "accumulated" risk measures. As far as I know, most of the RS-RL works name these "nested" versus "static" risk measures [1,2,3]. This confused me, as I could not understand the abstract until the middle of the introduction.

- line 62: "the optimal policy becomes history-dependent, which is more general than assuming the trajectory reward is a linear function of the sum of per-state features". Why? This deserves more explanation, as it justifies the novelty of this work compared to Chen et al (2023).

- Previous works have studied RL under trajectory feedback, although not in the PbRL setting, see e.g., [4]. Therefore, contribution 3. seems overstated.

[1] Hau, J. L., Petrik, M., & Ghavamzadeh, M. (2023, April). Entropic risk optimization in discounted MDPs. In International Conference on Artificial Intelligence and Statistics (pp. 47-76). PMLR.

[2] Hau, J. L., Delage, E., Ghavamzadeh, M., & Petrik, M. (2024). On dynamic programming decompositions of static risk measures in Markov decision processes. Advances in Neural Information Processing Systems, 36.

[3] Tamar, A., Chow, Y., Ghavamzadeh, M., & Mannor, S. (2015). Policy gradient for coherent risk measures. Advances in neural information processing systems, 28.

[4] Efroni, Y., Merlis, N., & Mannor, S. (2021, May). Reinforcement learning with trajectory feedback. In Proceedings of the AAAI conference on artificial intelligence (Vol. 35, No. 8, pp. 7288-7295).

**Questions:**

- The authors tackle nested and static CVaR with the same methodology. Classically, the static formulation encounters time consistency issues that do not appear in nested CVaR. Why isn't it the case here? Could the authors discuss time-consistent risks in the preference-based setup?

- What does Assp. 3.1 formally say? How should it be interpreted and how restrictive is it?

- lines 142-148: Why can we restrict policy-search to deterministic policies even in the preference-based setting? Is it an underlying assumption or is there a formal result stating that we do not lose optimality?

- Are the results from Sec. 3.2 novel - especially the recursions (5) and (6)?

- It is perhaps a philosophical concern but I would be glad to have the authors' feedback. If the feedback is preference-based, what is the difference between average and risk-sensitive return maximization? In other words, doesn't the agent's objective express itself directly through the preference outputs? This question is crucial to me because I currently do not see the motivation for RS-PbRL.

*Minor comments and suggestions*

- Please follow the formatting guidelines for section titles - Secs. 2, 5
- line 58: "may not be applicable"; line 60: "cannot"
- line 125: "and the reward"; line 126: "the trajectory embedding dimension"
- Assp. 3.1: "We" - capital letter
- line 133: " At each iteration"; line 134: "unlike standard RL"; line 137: "a Bernoulli distribution"
- lines 139-140: "It is aware" --> "It is known" or "We are aware that"; line 157: "risk-measures"
- line 160: "a random variable" (remove distribution); line 161: "then" --> "so"
- line 163: What is $Z$? Should it be $X$?
- line 176: "denotes the"
- lines 178-179: Reformulate "the policy should be history-dependent policies"
- line 185: "The proof is"
- Lemma 3.5 and 3.6: "For a tabular MDP and a reward of the entire trajectory being decomposable as ..."
- line 191: "given the current history"
- line 199: "The proof is"
- line 207: "We define"
- line 211: "regardless of iterated or accumulated"
- line 214: "of the entire episode training"
- line 218: "that minimizes regret"
- lines 221-22: "We present the algorithm called RA-PbRL and establish and upper-bound regret for it"
- line 225: Remove "The algorithm"; "defined" --> "described" or "presented"; "which main part"
- line 229: "choose a policy"; line 232: "find an estimated optimal policy"
- line 236: "we initialize"; line 237: "we observe history samples"; line 239: "use" (present)
- Algo 1, line 9: "Receive trajectory ... and preference"; line 10: italic $k$
- lines 253, 254: "with policy $\pi$" - remove "the"
- line 274: "hard-to-learn constructions" (hyphen and "s")
- lines 275-6: "are unavoidable in some cases"
- line 282: "For a comparative analysis"; line 286: "aligned ... with RA-PbRL"

**Limitations:**

Although some limitations are not addressed in this work (see previous remarks), most are properly described in Sec 6.

---

> ### Author Rebuttal · Authors · 2024-08-07
>
> Thanks for your insightful comments and recognizing the potential contribution of our work. Here are the responses to the issues you raised:
> # For Weakness1:
> We acknowledge your point. In fact, we find the terms "nested" and "static" risk measures are more widely accepted within the RS-RL community, as mentioned in your references [1,2,3]. We appreciate your critique and will revise our terminology.
>
> # For Weakness2:
> The reason the reward function is history-dependent and cannot be sum-decomposable is that, in many cases, the trajectory is embedded into a vector which is then dot-multiplied with a weight to obtain the reward（Pacchiano et al. (2021)）. (Details on the trajectory embedding can be found in line 124, under the Reward model of the entire trajectory section). Often, the embedding function is not even linear. Chen et al. (2023) represent a specific instance where the trajectory embedding is a linear embedding (see Remark 3.2). Our approach addresses two main limitations in Chen et al. (2023):
> 1. In many scenarios, it is not feasible to calculate the state-action reward function at each step. Consider a straightforward example where two policies each follow a determined  5 steps trajectory. Consequently, we can only obtain a preference equation and are unable to compute the 10 state-action reward function for each step.
>
> 2. In many application scenarios, we will only use trajectory embeddings. Practical and relevant trajectory or state-action embeddings are described in works such as Pacchiano et al. (2020) and Parker-Holder et al. (2020). Thus the embedded reward for a single episode may not always be simply represented as the sum of each state-action reward which is suggested by Chen et al. (2023).
> # For Weakness3:
> While prior studies have explored RL under trajectory feedback, our work distinctively focuses on risk-aware objectives, which fundamentally differ from previous approaches. Our algorithm specifically caters to both "nested" and "static" risk measures, The definition of value in our approach (as shown in Eq. 4 and 6, using quantile values) contrasts with the conventional mean value definition used in prior studies).
>
> # For Question1
>
> It is important to clarify that we do not employ the same methodology for two different objectives throughout. Although the algorithmic processes for calculating nested and static risk measures appear similar. Consequently, our value equations (Eq. 4 and 6) are distinct. Regarding the time-consistency issues associated with nested CVaR, we address these in Appendix F, which details the optimal policy calculation for known PbRL MDPs. This calculation is also applicable to Algorithm 1. We have expanded the state space to $\tilde{s}_h=\left(\xi_h, \rho\right)$, where $\rho$ functions as a quantile value. This expansion allows us to optimize iteratively by incorporating $\rho$, enhancing the algorithm’s efficacy in handling time-consistent risks.
> # For Question2:
> Formally, Assumption 3.1 implies that the absolute values of the non-zero components of trajectory embeddings have a lower bound b, or in other words, there is a gap between zero and some positive number b in the absolute values of components of trajectory embeddings. This is evident for finite-step discrete action spaces, where we can enumerate all trajectory embeddings to find the smallest non-zero component, satisfying most application scenarios. It is easy to demonstrate that B/b is always greater than or equal to the ratio of max reward to min reward. However, we can always perform a linear transformation on the embedding vector to make the equality hold, thus allowing us to use the ratio of max reward to min reward to estimate B/b.
>
> # For Question3:
> We postulate the existence of an optimal deterministic policy for any PbRL MDP, consistent with other PbRL research like Chen. et al, 2022.
> # For Question4:
> The recursions (5) and (6) introduced in Section 3.2 build upon existing methods from Du et al. (2022) and Pacchiano et al. (2023), adapted for our risk-aware PbRL context. These adaptations make our recursive formulas (4) and (6) novel, and specifically tailored to address the complexities of risk-sensitive value iteration.
> # For Question5:
> The agent expresses its objectives through preference outputs, specifically following a Bernoulli distribution `BN(p)` to choose the better trajectory, where `p` is determined by the reward. Essentially, our method infers the reward function and possibility kernel by observing the agent's objectives, thereby obtaining the policy's reward distribution. The distinction between average and risk-sensitive return maximization can be illustrated as follows:
> |          | Trajectory1                | Trajectory2                | CVaR(`\alpha=0.1`) |
> | -------- | -------------------------- | -------------------------- | -------------- |
> | Policy A | reward=0.1, possibility=0.1 | reward=0.5, possibility=0.9 | 0.1            |
> | Policy B | reward=0.2, possibility=0.1 | reward=0.3, possibility=0.9 | 0.2            |
>
> The agents favor Policy A (`P(A>B)=0.9`), but if we can infer and calculate the reward for each trajectory, we would find that Policy B is less risky (` B's CVaR  is lager`).  An example in real life might be: a large language model, fine-tuned with user feedback, generates two text options, A and B, for user selection. For most users, Text A is non-offensive and appears more comprehensive, thus preferred; however, Text A actually contains content offensive to a minority group. On average, the policy generating A performs better because it is chosen by the majority and only offends a few. Nonetheless, overall, we aim to avoid any offensive content completely, which is where risk sensitivity comes into play: after considering risk sensitivity, Text B would prevail. Compared to non-preference-based scenarios, preference-based feedback only affects our methods for estimating rewards and does not alter our consideration of risk.

---

> > ### Author Response · Authors · 2024-08-13
> >
> > We want to express our gratitude once again for your valuable feedback. Since the discussion period is coming to a close, we would be grateful if you could share any additional thoughts you may have about our rebuttal. Thank you for your time.

---

> > > ### Comment · Reviewer_TVSN · 2024-08-13
> > > **Thank you**
> > >
> > > Thank you for addressing all of my concerns and answering my questions. I  raise my score to 6

---

> > > > ### Author Response · Authors · 2024-08-13
> > > >
> > > > Thank you for acknowledging our rebuttal. We are glad to know that we addressed all your concerns.

---

### Author Rebuttal · Authors · 2024-08-07

We thank all the reviewers and ACs for their time and efforts in reviewing our paper and providing insightful comments. We acknowledge reviewers[TVSN,LP4v] for recognizing our contribution of applying both iterated and accumulated risk-aware to PbRL and reviewers [QEr2, EoFm]’s appreciating for our  mathematical development and theoretical analysis.  In addition to the detailed response to each reviewer, here we clarify common concerns and summarize the new results.
1. Since the experiment in our manuscript is simple, we implement our algorithm to solve MuJoCo's Half-cheetah simulation. This setting is more realistic than our previous toy setting and aims at learning how to control a simplified cross-section of a cheetah robot. Detailed information about the experiment can be found in the PDF. The results show our algorithm has a good performance compared to baseline, which demonstrates our algorithm and theory can guide real-life applications.
2. Risk-aware PbRL have many applications. For instance, in autonomous driving, PbRL can reduce the computational burden by obviating the need to calculate reward signals from every state-action pair, while incorporating risk-awareness to enhance safety (referenced in Chen et al., 2022). Additionally, in fields like generative AI, including systems like ChatGPT (discussed in Zhuo et al., 2023; Qi et al., 2023; Chen et al., 2023), PbRL is utilized to gauge algorithm performance where managing risk is crucial to prevent the generation of harmful content. We agree that emphasizing these applications could better highlight the importance of our contributions.

---

### Decision · Program_Chairs · 2024-09-25

**Decision:**

Accept (poster)

**Comment:**

Preference-based reinforcement learning has been investigate under its full settings and reduced formats (preference-based optimization). This paper investigates its conservative form with theoretical guarantees.

Traditional PbRL aims to maximize mean reward or utility, which is a risk-neutral approach. The proposed RA-PbRL incorporates risk-aware objectives to address scenarios where risk sensitivity is crucial, e.g. in healthcare. The theoretical analysis of RA-PbRL includes establishing regret upper bounds that are sublinear w.r.t.. the number of episodes. The empirical analysis shows that the regret of the RA-PbRL algorithm is lower than that of other representative algorithms.

The meta reviewer read the paper and discussions. Although there is a lack of references on preference-based RL and risk-aware RL, this paper has reached the bar for NeurIPS. Reviewer LP4v considers the paper was written hurriedly. The AC agrees with Reviewer LP4v that this paper can benefit from a detailed round of revision before the camera ready submission.